# INTENTION PROPAGATION FOR MULTI-AGENT REINFORCEMENT LEARNING

## ABSTRACT

A hallmark of an AI agent is to mimic human beings to understand and interact with others. In this paper, we propose a *collaborative* multi-agent reinforcement learning algorithm to learn a *joint* policy through the interactions over agents. To make a joint decision over the group, each agent makes an initial decision and tells its policy to its neighbors. Then each agent modifies its own policy properly based on received messages and spreads out its plan. As this intention propagation procedure goes on, we prove that it converges to a mean-field approximation of the joint policy with the framework of neural embedded probabilistic inference. We evaluate our algorithm on several large scale challenging tasks and demonstrate that it outperforms previous state-of-the-arts.

## 1 INTRODUCTION

Collaborative multi-agent reinforcement learning is an important sub-field of the multi-agent reinforcement learning (MARL), where the agents learn to coordinate to achieve joint success. It has wide applications in traffic control (Kuyer et al., 2008), autonomous driving (Shalev-Shwartz et al., 2016) and smart grid (Yang et al., 2018). To learn a coordination, the interactions between agents are indispensable. For instance, humans can reason about other's behaviors or know other peoples' intentions through communication and then determine an effective coordination plan. However, how to design a mechanism of such interaction in a *principled* way and at the same time solve the large scale real-world applications is still a challenging problem.

Recently, there is a surge of interest in solving the collaborative MARL problem (Foerster et al., 2018; Qu et al., 2019; Lowe et al., 2017). Among them, joint policy approaches have demonstrated their superiority (Rashid et al., 2018; Sunehag et al., 2018; Oliehoek et al., 2016). A straightforward approach is to replace the action in the single-agent reinforcement learning by the joint action $\mathbf{a} = (a_1, a_2, ..., a_N)$, while it obviously suffers from the issue of the exponentially large action space. Thus several approaches have been proposed to factorize the joint action space to mitigate such issue, which can be roughly grouped into two categories:

- Factorization on policy. This approach explicitly assumes that $\pi(\mathbf{a}|s) := \prod_{i=1}^{N} \pi_i(a_i|s)$, i.e., policies are independent (Foerster et al., 2018; Zhang et al., 2018). To mitigate for the instability issue caused by the independent learner, it generally needs a centralized critic.
- Factorization on value function. This approach has a similar spirit but factorizes the joint value function into several utility functions, each just involving the actions of *one agent* (Rashid et al., 2018; Sunehag et al., 2018).

However, these two approaches *lack of the interactions* between agents, since in their algorithms agent $i$ does not care about the plan of agent $j$. Indeed, they may suffer from a phenomenon called relative over-generalization in game theory observed by Wei & Luke (2016); Castellini et al. (2019); Palmer et al. (2018). Approaches based on the coordinate graph would effectively prevent such cases, where the value function is factorized as a summation of utility function on pairwise or local joint action (Guestrin et al., 2002; Böhmer et al., 2020). However, they only can be applied in discrete action, small scale game.

Furthermore, despite the empirical success of the aforementioned work in certain scenarios, it still *lacks* theoretical insight. In this work, we only make a simple yet realistic assumption: the reward function $r_i$ of each agent $i$ just depends on its individual action and the actions of its neighbors (and

state), i.e.,

$$r_i(s, \mathbf{a}) = r_i(s, a_i, a_{\mathcal{N}_i}), \tag{1}$$

where we use $\mathcal{N}_i$ to denote the neighbors of agent $i$, $s$ to denote the global state. It says the goal or decision of agent is explicitly influenced by a small subset $\mathcal{N}_i$ of other agents. Note that such an assumption is *reasonable in lots of real scenarios*. For instance,

- The traffic light at an intersection makes the decision on the phase changing mainly relying on the traffic flow around it and the policies of its neighboring traffic light.
- The main goal of a defender in a soccer game is to tackle the opponent's attacker, while he rarely needs to pay attention to opponent goalkeeper's strategy.

Based on the assumption in equation 1, we propose a *principled* multi-agent reinforcement learning algorithm in the framework of probabilistic inference, where the objective is to maximize the long term reward of the group, i.e., $\sum_{t=0}^{\infty} \sum_{i=1}^{N} \gamma^t r_i^t$ ( see details in section 4).

Note since each agent's reward depends on its neighbor, we still need a joint policy to maximize the global reward through *interactions*. In this paper, we derive an iterative procedure for such interaction to learn the joint policy in collaborative MARL and name it *intention propagation*. Particularly,

- In the first round, each agent $i$ makes an independent decision and spreads out his plan $\tilde{\mu}_i$(we name it *intention*) to neighbors.
- In the second round, agents $i$ changes its initial intention *properly* based on its neighbors' intention $\tilde{\mu}_j, j \in \mathcal{N}_i$ and propagates its intention $\tilde{\mu}_i$ again.
- In the third round, it changes the decision in the second round with a similar argument.
- As this procedure goes on, we show the final output of agents' policy converges to the mean field approximation (the variational inference method from the probabilistic graphical model (Bishop, 2006)) of the joint policy.

In addition, this joint policy has the form of Markov Random Field induced by the locality of the reward function (proposition 1). Therefore, such a procedure is computationally efficient when the underlying graph is sparse, since in each round, each agent just needs to care about what its neighbors intend to do. **Remark:** (1) Our work is not related to the *mean-field game* (MFG) (Yang et al., 2018). The goal of the MFG is to find the Nash equilibrium, while our work aims to the optimal joint policy in the *collaborative* game. Furthermore, MFG generally assumes agents are identical and interchangeable. When the number of agents goes to infinity, MFG can view the state of other agents as a population state distribution. In our problem, we do not have such assumptions. (2) our analysis is not limited to the mean-field approximation. When we change the message passing structure of intention propagation, we can show that it converges to other approximation of the joint policy, e.g., *loopy belief propagation* in variational inference (Yedidia et al., 2001) (see **Appendix** B.2 ).

**Contributions:** (1) We propose a *principled* method named intention propagation to solve the joint policy collaborative MARL problem; (2) Our method is computationally efficient, which can scale up to **one thousand** agents and thus meets the requirement of real applications; (3) Empirically, it outperforms state-of-the-art baselines with a wide margin when the number of agents is large; (4) Our work builds a bridge between MARL and neural embedded probabilistic inference, which would lead to new algorithms beyond intention propagation.

**Notation**: $s_i^t$ and $a_i^t$ represent the state and action of agent $i$ at time step $t$. The neighbors of agent $i$ are represented as $\mathcal{N}_i$. We denote $X$ as a random variable with domain $\mathcal{X}$ and refer to instantiations of $X$ by the lower case character $x$. We denote a density on $\mathcal{X}$ by $p(x)$ and denote the space of all such densities as by $\mathcal{P}$.

## 2 RELATED WORK

We first discuss the work of the factorized approaches on the joint policy. COMA designs a MARL algorithm based on the actor-critic framework with independent actors $\pi_i(a_i|s)$, where the joint policy is factorized as $\pi(\mathbf{a}|s) = \prod_{i=1}^{N} \pi_i(a_i|s)$ (Foerster et al., 2018). MADDPG considers a MARL with the cooperative or competitive setting, where it creates a critic for each agent (Lowe et al., 2017). Other similar works may include (de Witt et al., 2019; Wei et al., 2018). Another way is to factorize the value functions into several utility functions. Sunehag et al. (2018) assumes that the

overall Q function can be factorized as $Q(s, a_1, a_2, .., a_N) = \sum_{i=1}^{N} Q_i(s_i, a_i)$ . QMIX extends this work to include a richer class of function, where it assumes the overall $Q$ function is a monotonic function w.r.t. each $Q_i(s_i, a_i)$ (Rashid et al., 2018). Similarly, Son et al. (2019) further relax the structure constraint on the joint value function. However these factorized methods suffer from the relative overgeneralization issue (Castellini et al., 2019; Palmer et al., 2018). Generally speaking, it pushes the agents to underestimate a certain action because of the low rewards they receive, while they could get a higher one by perfectly coordinating.

A middle ground between the (fully) joint policy and the factorized policy is the coordination graph (Guestrin et al., 2002), where the value function is factorized as a summation of the utility function on the pairwise action. Böhmer et al. (2020); Castellini et al. (2019) combine deep learning techniques with the coordination graph. It addresses the issue of relative overgeneralization, but still has two limitations especially in the large scale MARL problem. (1) The max-sum algorithm can just be implemented in the discrete action space since it needs a max-sum operation on the action of $Q$ function. (2) Even in the discrete action case, *each step* of the Q learning has to do several loops of max-sum operation over the whole graph if there is a cycle in graph. Our algorithm can handle both discrete and continuous action space cases and alleviate the scalability issue by designing an intention propagation network.

Another category of MARL is to consider the communication among agents. The attention mechanism is used to decide when and who to communicate with (Das et al., 2018). Foerster et al. (2016) propose an end-to-end method to learn communication protocol. In (Liu et al., 2019; Chu et al., 2020), each agent sends the action information to it neighbors. In addition, Chu et al. (2020) require a strong assumption that the MDP has the spatial-temporal Markov property. However, they utilizes neighbor's action information in a heuristic way and thus it is unclear what the agents are learning (e.g., do they learn the optimal joint policy to maximize the group reward? ). Jiang et al. (2020) propose DGN which uses GNN to spread the *state* embedding information to neighbors. However each agent still uses an independent Q learning to learn the policy and neglects other agents' plans. In contrast, we propose a principled algorithm, where each agent makes decision considering other agents' plan. Such procedure can be parameterized by GNN and other neural networks (see section 4.1 and appendix B.2). We prove its convergence to the solution of variational inference methods.

## 3 BACKGROUNDS

**Probabilistic Reinforcement Learning:** Probabilistic reinforcement learning (PRL) (Levine, 2018) is our building block. PRL defines the trajectory $\tau$ up to time step $T$ as $\tau = [s^0, a^0, s^1, a^1, ..., s^T, a^T, s^{T+1}]$. The probability distribution of the trajectory $\tau$ induced by the *optimal policy* is defined as $p(\tau) = [p(s^0) \prod_{t=0}^{T} p(s^{t+1}|s^t, a^t)] \exp\left(\sum_{t=0}^{T} r(s^t, a^t)\right)$. While the probability of the trajectory $\tau$ under the policy $\pi(a|s)$ is defined as $\hat{p}(\tau) = p(s^0) \prod_{t=0}^{T} p(s^{t+1}|s^t, a^t)\pi(a^t|s^t)$. The objective is to minimize the KL divergence between $\hat{p}(\tau)$ and $p(\tau)$. It is equivalent to the maximum entropy reinforcement learning

$$\max_{\pi} J(\pi) = \sum_{t=0}^{T} \mathbb{E}[r(s^t, a^t) + \mathcal{H}(\pi(a^t|s^t))],$$

where it omits the discount factor $\gamma$ and regularizer factor $\alpha$ of the entropy term, since it is easy to incorporate them into the transition and reward respectively. Notice in this framework the max operator in the Bellman optimality equation would be replaced by the softmax operator and thus its optimal policy is a softmax function related to the Q function (Haarnoja et al., 2017). Such framework subsumes state-of-the-art algorithms such as soft-actor-critic (SAC) (Haarnoja et al., 2018). In each iteration, SAC optimizes the following loss function of $Q, \pi, V$, and respectively.

$$\mathbb{E}_{(s^t,a^t)\sim D}\left[Q(s^t, a^t) - r(s^t, a^t) - \gamma\mathbb{E}_{s^{t+1}\sim p}[V(s^{t+1})]\right]^2, \mathbb{E}_{s^t\sim D}\mathbb{E}_{a^t\sim\pi}[\log \pi(a^t|s^t) - Q(s^t, a^t)]$$

$$\mathbb{E}_{s^t\sim D}\left[V(s^t) - \mathbb{E}_{a^t\sim\pi_\theta}[Q(s^t, a^t) - \log \pi(a^t|s^t)]\right]^2, \text{ where } D \text{ is the replay buffer.}$$

**Function Space Embedding of Distribution:** In our work, we use the tool of embedding in Reproducing Kernel Hilbert Space (RKHS) to design an intention propagation procedure (Smola et al., 2007). We let $\phi(X)$ be an implicit feature mapping and $X$ be a random variable with distribution $p(x)$. Embeddings of $p(x)$ is given by $\mu_X := \mathbb{E}_X[\phi(X)] = \int \phi(x)p(x)dx$ where the distribution is mapped to its expected feature map. By assuming that there exists a feature space such that

the embeddings are injective, we can treat the embedding $\mu_X$ of the density $p(x)$ as a sufficient statistic of the density, i.e., any information we need from the density is preserved in $\mu_X$ (Smola et al., 2007). Such injective assumption generally holds under mild condition (Sriperumbudur et al., 2008). This property is important since we can reformulate a functional $f : \mathcal{P} \to \mathbb{R}$ of $p(\cdot)$ using the embedding only, i.e., $f(p(x)) = \tilde{f}(\mu_X)$. It also can be generalized to the operator case. In particular, applying an operator $\mathcal{T} : \mathcal{P} \to \mathbb{R}^d$ to a density can be equivalently carried out using its embedding $\mathcal{T} \circ p(x) = \tilde{\mathcal{T}} \circ \mu_X$, where $\tilde{\mathcal{T}} : \mathcal{F} \to \mathbb{R}^d$ is the alternative operator working on the embedding. In practice, $\mu_X$, $\tilde{f}$ and $\tilde{\mathcal{T}}$ have complicated dependence on $\phi$. As such, we approximate them by neural networks, which is known as the neural embedding approach of distribution (Dai et al., 2016).

## 4 OUR METHOD

In this section, we present our method intention propagation for the collaborative multi-agent reinforcement learning. To begin with, we formally define the problem as a networked MDP. The network is characterized by a graph $\mathcal{G} = (\mathcal{V}, \mathcal{E})$, where each vertex $i \in \mathcal{V}$ represents an agent and the edge $ij \in \mathcal{E}$ means the communication link between agent $i$ and $j$. We say $i,j$ are neighbors if they are connected by this edge. The corresponding networked MDP is characterized by a tuple $(\{\mathcal{S}_i\}_{i=1}^N, \{\mathcal{A}_i\}_{i=1}^N, p, \{r_i\}_{i=1}^N, \gamma, \mathcal{G})$, where $N$ is the number of agents, $S_i$ is the local state of the agent $i$ and $\mathcal{A}_i$ denotes the set of action available to agent $i$. We let $S := \prod_{i=1}^N S_i$ and $\mathcal{A} := \prod_{i=1}^N \mathcal{A}_i$ be the global state and joint action space respectively. At time step $t+1$, the global state $s^{t+1} \in S$ is drawn from the transition $s^{t+1} \sim p(\cdot|s^t, \mathbf{a^t})$, conditioned on the current state $s^t$ and the joint action $\mathbf{a}^t = (a_1^t, a_2^t, ..., a_N^t) \in \mathcal{A}$. Each transition yields a reward $r_i^t = r_i(s^t, \mathbf{a^t})$ for agent $i$ and $\gamma$ is the discount factor. The aim of our algorithm is to learn a joint policy $\pi(\mathbf{a^t}|s^t)$ to maximize the overall long term reward (with an entropy term $\mathcal{H}(\cdot|s)$ on the joint action $\mathbf{a}$)

$$\eta(\pi) = \mathbb{E}[\sum_{t=0}^{\infty} \gamma^t (\sum_{i=1}^N r_i^t + \mathcal{H}(\cdot|s^t))],$$

where each agent $i$ can just observe its own state $s_i$ and the message from the neighborhood communication. We denote the neighbors of agent $i$ as $\mathcal{N}_i$ and further assume that the reward $r_i$ depends on the state and the actions of itself and its neighbors, i.e., $r_i(s, \mathbf{a}) := r_i(s, a_i, a_{\mathcal{N}_i})$. Such assumption is reasonable in many real scenarios as we discussed in the introduction. In the following, we start the derivation with the fully observation case, and discuss how to handle the partial observation later. The roadmap of the following derivation : At the beginning, we prove that the optimal policy has a Markov Random Field (MRF) form, which reduces the exponential large searching space to a polynomial one. However implement a MRF policy is not trivial in the RL setting (e.g., sample an action from the policy). Thus we sort to the varational inference method (focus on mean field approximation in the main paper and leave other methods in the appendix). But it would introduce complicated computations. At last we apply the kernel embedding method introduced in section 3 to solve this problem and learn the kernel embedding by neural networks. We also discuss how to handle the partially observable setting.

### 4.1 REDUCE POLICY SEARCHING SPACE

Recall that our aim is to maximize the long term reward with the entropy term. Therefore, we follow the definition of the optimal policy in the probabilistic reinforcement learning in (Levine, 2018) and obtain the proposition 1. It says under the assumption $r_i(s, a) = r_i(s, a_i, a_{\mathcal{N}_i})$, the optimal policy is in the form of Markov Random Field (MRF). We prove the following proposition in I.1.

**Proposition 1** *The optimal policy has the form* $\pi^*(\mathbf{a^t}|s^t) = \frac{1}{Z} \exp(\sum_{i=1}^N \psi_i(s^t, a_i^t, a_{\mathcal{N}_i}^t))$, *where* $Z$ *is the normalization term.*

This proposition is important since it suggests that we should construct the policy $\pi(\mathbf{a^t}|s^t)$ with this form, e.g., a parametric family, to contain the optimal policy. If agent $i$ and its neighbors compose a clique, the policy reduces to a MRF and $\psi$ is the potential function. One common example is that the reward is a function on pairwise actions, i.e., $r(s, \mathbf{a}) = \sum_{i \in \mathcal{V}} r(s, a_i) + \sum_{(i,j) \in \mathcal{E}} r(s, a_i, a_j)$. Then the policy has the form

$$\pi(\mathbf{a}|s) = \frac{1}{Z} \exp(\sum_{i \in \mathcal{V}} \tilde{\psi}_i(s, a_i) + \sum_{(i,j) \in \mathcal{E}} \tilde{\psi}_{i,j}(s, a_i, a_j)),$$

which is the pairwise MRF. For instance, in traffic lights control, we can define a 2-D grid network and the pairwise reward function. The MRF formulation on the policy effectively reduces the policy space comparing with the exponentially large one in the fully connected graph.

A straightforward way to leverage such observation is to define a $\pi_\theta(\mathbf{a}^t|s^t)$ as a MRF, and then apply the policy gradient algorithm, e.g., the following way in SAC. $\nabla_\theta \mathbb{E}_{s^t \sim D} \mathbb{E}_{a^t \sim \pi_\theta}[\log \pi_\theta(\mathbf{a}^t|s^t) - Q_\kappa(s^t, \mathbf{a}^t)]$. However it is still very *hard* to sample joint action $\mathbf{a}^t$ from $\pi_\theta(\mathbf{a}^t|s^t)$. In the next section, we resort to embedding to alleviate such problem.

Recall the remaining problem is how to sample the joint action from a MRF policy. Classical ways include the Markov Chain Monte Carlo method and *variational inference*. The former provides the guarantee of producing exact samples from the target density but computationally intensive. Therefore it is not applicable in the multi-agent RL setting, since we need to sample action once in each interaction with the environment. As such, we advocate the second approach. Here we use the mean-field approximation for the simplicity of presentation and defer more variational inference methods, e.g., loopy belief propagation, in **Appendix** B.2. We use an intention propagation network with the embedding of the distribution to represent the update rule of the mean field approximation.

**Mean field approximation.** We hope to approximate the $\pi^*(a|s)$ by the mean-field variational family $p_i$

$$\min_{(p_1, p_2, \dots, p_N)} KL(\prod_{i=1}^{N} p_i(a_i|s) || \pi^*(\mathbf{a}|s)),$$

where we omit the superscript $t$ to simplify the notation. We denote the optimal solution of above problem as $q_i$. Using the coordinate ascent variational inference,the optimal solution $q_i$ should satisfy the following fixed point equation (Bishop, 2006). Since the objective function is (generally) non-convex, such update converges to a local optimum (Blei et al., 2017).

$$q_i(a_i|s) \propto \exp \int \prod_{j \neq i} q_j(a_j|s) \log \pi^*(\mathbf{a}|s) d\mathbf{a}. \tag{2}$$

For simplicity of the representation, in the following discussion, we assume that the policy is a pairwise MRF but the methodology applies to more general case with more involved expression. Particularly, we assume $\pi^*(\mathbf{a}|s) = \frac{1}{Z} \exp(\sum_{i \in \mathcal{V}} \psi_i(s, a_i) + \sum_{(i,j) \in \mathcal{E}} \psi_{ij}(s, a_i, a_j))$. We plug this into equation 2 and obtain following fixed point equation.

$$\log q_i(a_i|s) = c_i + \psi_i(s, a_i) + \sum_{j \in \mathcal{N}_i} \int q_j(a_j|s) \psi_{ij}(s, a_i, a_j) da_j, \tag{3}$$

where $c_i$ is some constant that does not depend on $a_i$.

We can understand this mean-field update rule from the perspective of intention propagation. Equation 3 basically says each agent can *not* make the decision independently. Instead its policy $q_i$ should depend on the policies of others, particularly the neighbors in the equation. Clearly, if we can construct the intention propagation corresponding to equation 3, the final policy obtained from intention propagation will *converge* to the mean-field approximation of the joint policy. However we can *not* directly apply this update in our algorithm, since it includes a complicated integral. To this end , in the next section   we resort to the embedding of the distribution $q_i$ (Smola et al., 2007) , which maps the distributions into a reproducing kernel Hilbert space.

**Embed the update rule.**   Observe that the fixed point formulation equation 3 says that $q_i(a_i|s)$ is a functional of neighborhood marginal distribution $\{q_j(a_j|s)\}_{j \in \mathcal{N}_i}$, i.e., $q_i(a_i|s) = f(a_i, s, \{q_j\}_{j \in \mathcal{N}_i})$. Denote the d-dimensinoal embedding of $q_j(a_j|s)$ by $\tilde{\mu}_j = \int q_j(a_j|s) \phi(a_j|s) da_j$. Notice the form of feature $\phi$ is not fixed at the moment and will be learned implicitly by the neural network. Following the assumption that there exists a feature space such that the embeddings are injective in Section 3, we can replace the distribution by its embedding and have the fixed point formulation as

$$q_i(a_i|s) = \tilde{f}(a_i, s, \{\tilde{\mu}_j\}_{j \in \mathcal{N}_i}). \tag{4}$$

For more theoretical guarantee on the kernel embedding, e.g., convergence rate on the empirical mean of the kernel embedding, please refer to (Smola et al., 2007). Roughly speaking, once there

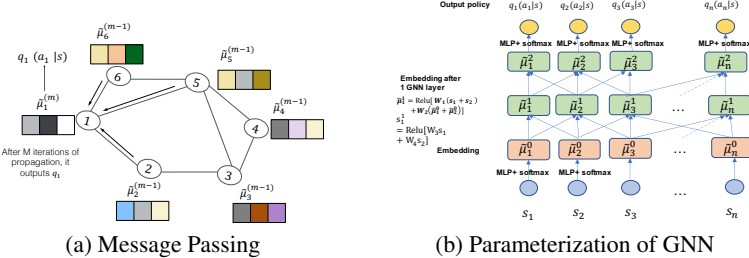

(a) Message Passing       (b) Parameterization of GNN

Figure 1: (a) Illustration of the message passing of intention propagation $\Lambda_\theta(\mathbf{a}|s)$ ( equation 5). (b) An instance of 2-layer GNN with the discrete action outputs (n agents).

are enough data, we can believe the learned kernel embedding is close enough to the true kernel embedding. Therefore the update of equation 4 and equation 5 in the following would converge to the fixed point of equation 2. Remind that in section 3 at both sides we can do integration w.r.t. the feature map $\phi$, which yields, $\tilde{\mu}_i = \int q_i(a_i|s)\phi(a_i|s)da_i = \int \tilde{f}(a_i, s, \{\tilde{\mu}_j\}_{j \in \mathcal{N}_i})\phi(a_i|s)da_i$. Thus we can rewrite it as a new operator on the embedding, which induces a fixed point equation again $\tilde{\mu}_i = \tilde{\mathcal{T}} \circ (s, \{\tilde{\mu}_j\}_{j \in \mathcal{N}_i})$. In practice, we do this fix-point update with $M$ iterations.

$$\tilde{\mu}_i^m \leftarrow \tilde{\mathcal{T}} \circ (s, \{\tilde{\mu}_j^{m-1}\}_{j \in \mathcal{N}_i}) \quad m = 1, ..., M. \tag{5}$$

Finally, we output the distribution $q_i$ with $q_i(a_i|s) = \tilde{f}(a_i, s, \{\tilde{\mu}_j^M\}_{j \in \mathcal{N}_i})$. In next section, we show how to represent these variables by neural network.

**Parameterization by Neural Networks.** In general $\tilde{f}$ and $\tilde{\mathcal{T}}$ have complicated dependency on $\psi$ and $\phi$. Instead of learning such dependency, we directly approximate $\tilde{f}$ and $\tilde{T}$ by neural networks. For instance, we can represent the operator $\tilde{\mathcal{T}}$ in equation 5 by $\tilde{\mu}_i = \sigma(W_1 s + W_2 \sum_{j \in \mathcal{N}_i} \tilde{\mu}_j)$, where $\sigma$ is a nonlinear activation function, $W_1$ and $W_2$ are some matrixes with row number equals to $d$. Interestingly, this is indeed a message passing form of *Graph Neural Network* (GNN) (Hamilton et al., 2017). Thus we can use $M$-hop (layer) GNN to represent the fixed-point update in equation 5. If the action space is discrete, the output $q_i(a_i|s)$ is a softmax function. In this case $\tilde{f}$ is a fully connected layer with softmax output. When it is continuous, we can output a Gaussian distribution with the reparametrization trick (Kingma & Welling, 2019). We denote this intention propagation procedure as intention propagation network $\Lambda_\theta(\mathbf{a}|s)$ with parameter $\theta$ in Figure 1(b).

Figure 1(a) illustrates the graph and the message passing procedure. Agent 1 receives the embedding (intention) $\tilde{\mu}_2^{m-1}, \tilde{\mu}_5^{m-1}, \tilde{\mu}_6^{m-1}$ from its neighbors, and then updates the its own embedding with operator $\tilde{\mathcal{T}}$ and spreads its new embedding $\tilde{\mu}_1^m$ at the next iteration. Figure 1(b) gives the details on the parameterization of GNN. Here we use agent 1 as an example. To ease the exposition, we assume agent 1 just has one neighbor, agent 2. Each agent observes its own state $s_i$. After a MLP and softmax layer (we do not sample actions here, but just use the probabilities of the actions), we get a embedding $\tilde{\mu}_i^0$, which is the initial distribution of the policy. Then agent 1 receives the embedding $\tilde{\mu}_2^0$ of its neighbor (agent 2). After a GNN layer to combine the information, e.g, $\tilde{\mu}_1^1 = Relu[W_1(s_1 + s_2) + W_2(\tilde{\mu}_1^0 + \tilde{\mu}_2^0)](W_1, W_2$ are shared across all agents as that in GNN), we obtain new embedding $\tilde{\mu}_1^1$ of agent 1. Notice we also do message passing on state, since in practice the global state is not available. In the second layer, we do similar things. We defer detailed *discussion and extension to other neural networks* to **Appendix** B due to space constraint.

### 4.2 ALGORITHM

We are ready to give the overall algorithm by combining all pieces together. All detailed derivation on $V_i, Q_i$ for agent $i$ and the corresponding loss function will be given in the appendix I, due to the space constraint. Recall we have a mean-field approximation $q_i$ of the joint-policy, which is obtained by $M$ iterations of intention propagation. We represent this procedure by a M-hop graph neural network with parameter $\theta$ discussed above. Notice that this factorization is *different* from the case $\pi(\mathbf{a}|s) = \prod_{i=1}^N \pi(a_i|s)$ in (Zhang et al., 2018; Foerster et al., 2018), since $q_i(a_i|s)$ depends on the information of other agents' plan. Using the mean field approximation $q_i$, we can further decompose $Q = \sum_{i=1}^N Q_i$ and $V = \sum_{i=1}^N V_i$, see appendix I. We use neural networks to approximate $V_i$ and $Q_i$ function with parameter $\eta_i$ and $\kappa_i$ respectively. As that in TD3 (Fujimoto et al., 2018), for each agent $i$ we have a target value network $V_{\bar{\eta}_i}$ and two $Q_{\kappa_i}$ functions to mitigate the overestimation by training them simultaneously with the same data but only selecting minimum of them as the

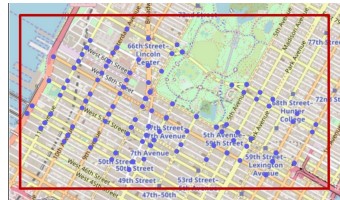 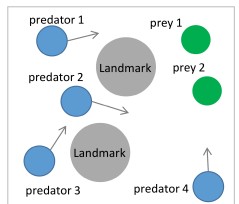 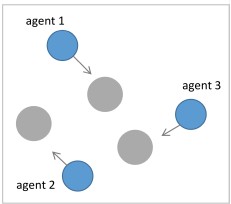

Figure 2: Experimental scenarios. Cityflow: Manhattan, Predator-Prey and Cooperative-Navigation.

target in the value update. In the following we denote $q_i(a_i|s)$ as $q_{i,\theta}(a_i|s)$ to explicitly indicate its dependence on the intention propagation network $\Lambda_\theta$. We use $D$ to denote the replay buffer. The whole algorithm is presented in Algorithm 1.

**Loss Functions.** The loss of value function $V_i$:

$$J(\eta_i) = \mathbb{E}_{s^t \sim D}[\frac{1}{2}\big(V_{\eta_i}(s^t) - \mathbb{E}_{(a_i^t, a_{\mathcal{N}_i}^t) \sim (q_i, q_{\mathcal{N}_i})}[Q_{\kappa_i}(s^t, a_i^t, a_{\mathcal{N}_i}^t) - \log q_{i,\theta}(a_i^t|s^t)]\big)^2].$$

The loss of $Q_i$: $J(\kappa_i) = \mathbb{E}_{(s^t, a_i^t, a_{\mathcal{N}_i}^t) \sim D}[\frac{1}{2}\big(Q_{\kappa_i}(s^t, a_i^t, a_{\mathcal{N}_i}^t) - \hat{Q}_i(s^t, a_i^t, a_{\mathcal{N}_i}^t)\big)^2],$

where $\hat{Q}_i(s^t, a_i^t, a_{\mathcal{N}_i}^t) = r_i + \gamma \mathbb{E}_{s^{t+1} \sim p(\cdot|s_t, \mathbf{a}^t)}[V_{\bar{\eta}_i}(s^{t+1})].$

The loss of policy: $J(\theta) = \mathbb{E}_{s^t \sim D, \mathbf{a}^t \sim \prod_{i=1}^N q_i}[\sum_{i=1}^N \log q_{i,\theta}(a_i^t|s^t) - \sum_{i=1}^N Q_{\kappa_i}(s^t, a_i^t, a_{\mathcal{N}_i}^t)].$

It is interesting to compare the loss with the counterpart in the single agent SAC in section 3.

- $q_{i,\theta}(a_i|s)$ is the output of intention propagation network $\Lambda_\theta(\mathbf{a}|s)$ parameterized by a graph neural network. Thus it depends on the policy of other agents.
- $Q_{\kappa_i}$ depends on the action of itself and its neighbors, which can also be accomplished by the graph neural network in practice.

---

**Algorithm 1** Intention Propagation

**Inputs**: Replay buffer $D$. $V_i, Q_i$ for each agent $i$. Intention propagation network $\Lambda_\theta(\mathbf{a}_t|s)$ with outputs $\{q_{i,\theta}\}_{i=1}^N$. Learning rate $l_\eta, l_\kappa, l_\theta$. Moving average parameter $\tau$ for the target network
**for** each iteration **do**
  **for** each environment step **do**
    sample $\mathbf{a_t} \sim \prod q_{i,\theta}(a_i^t|s^t)$ from the intention propagation network. $s^{t+1} \sim p(s^{t+1}|s^t, \mathbf{a}^t)$,
    $D \leftarrow D \bigcup \big(s_i^t, a_i^t, r_i^t, s_i^{t+1}\big)_{i=1}^N$
  **end for**
  **for** each gradient step **do**
    update $\eta_i, \kappa_i, \theta, \bar{\eta}_i$.
$$\eta_i \leftarrow \eta_i - l_\eta \nabla J(\eta_i), \kappa_i \leftarrow \kappa_i - l_\kappa \nabla J(\kappa_i)$$
$$\theta \leftarrow \theta - l_\theta \nabla J(\theta), \bar{\eta}_i \leftarrow \tau \eta_i + (1-\tau)\bar{\eta}_i$$
  **end for**
**end for**

---

**Handle the Partial Observation:** So far, we assume that agents can observe global state while in practice, each agent just observes its own state $s_i$. Thus besides the communication with the intention propagation, we also do the message passing on the state embedding with the graph neural network. The idea of this local state sharing is similar to (Jiang et al., 2020), while the whole structure of our work is quite different from (Jiang et al., 2020). See the discussion in the related work.

## 5 EXPERIMENT

In this section, we evaluate our method and eight state-of-the-art baselines on more than ten different scenarios from three popular MARL platforms: (1) CityFlow, a traffic signal control environment

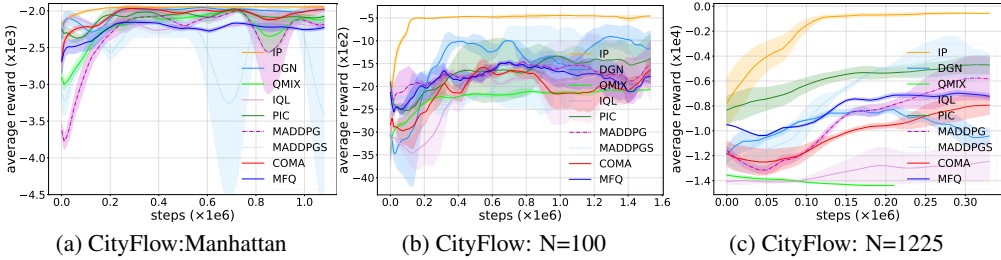

(a) CityFlow:Manhattan      (b) CityFlow: N=100      (c) CityFlow: N=1225

Figure 3: Performance on large-scale traffic lights control scenarios in CityFlow. Horizontal axis: environmental steps. Vertical axis: average episode reward (negative average travel time). Higher rewards are better. Our intention propagation (IP) performs best especially on large-scale tasks.

(Tang et al., 2019). It is an advanced version of SUMO (Lopez et al., 2018) widely used in MARL community. (2) multiple particle environment (MPE) (Mordatch & Abbeel, 2017) and (3) grid-world platform MAgent (Zheng et al., 2018). Our intention propagation (IP) empirically outperforms all baselines on all scenarios especially on the *large scale* problem.

## 5.1 SETTINGS

We give a brief introduction to the settings of the experiment and defer the details such as hyper-parameter tuning of intention propagation and baselines to appendix D. Notice all algorithms are tested in the partially observable setting, i.e., each agent just can observe its own state $s_i$.

In traffic signal control problem (Left panel in Figure 2), each traffic light at the intersection is an agent. The goal is to learn policies of traffic lights to reduce the average waiting time to alleviate the traffic jam. *Graph for cityflow*: graph is a 2-D grid induced by the map (e.g. Figure 2). The roads are the edges which connects the agents. We can define the cost $-r_i$ as the traveling time of vehicle around the intersection $i$, thus the total cost indicates the average traveling time. Obviously, $r_i$ has a close relationship with the action of neighbors of agent $i$ but has little dependence on the traffic lights far away. Therefore our assumption on reward function holds. We evaluate different methods on both real-world and synthetic traffic data under the different numbers of intersections.

MPE (Mordatch & Abbeel, 2017) and MAgent (Zheng et al., 2018) (Figure 2) are popular particle environments on MARL (Lowe et al., 2017; Jiang et al., 2020). *Graph for particle environments* : for each agent, it has connections (i.e., the edge of the graph) with $k$ nearest neighbors. Since the graph is dynamic, we update the adjacency matrix of the graph every $n$ step, e.g., $n = 5$ steps. It is just a small overhead comparing with the training of the neural networks. The reward functions also have local property, since they are explicitly or implicitly affected by the distance between agents. For instance, in heterogeneous navigation, if small agents collide with big agents, they will obtain a large negative reward. Thus their reward depends on the action of the nearby agents. Similarly, in the jungle environment, agent can attack the agents nearby to obtain a high reward.

**Baselines.** We compare our method against eight different baselines mentioned in introduction and related work section: QMIX (Rashid et al., 2018); MADDPG (Lowe et al., 2017); permutation invariant critic (PIC) (Liu et al., 2019); graph convolutional reinforcement learning (DGN) (Jiang et al., 2020); independent Q-learning (IQL) (Tan, 1993); permutation invariant MADDPG with data shuffling mechanism (MADDPGS); COMA (Foerster et al., 2018); MFQ (Yang et al., 2018). These baselines are reported as the leading algorithm of solving tasks in CityFlow, MPE and MAgent. Among them, DGN and MFQ need the communication with neighbors in the training and execution. Also notice that PIC assumes the actor can observe the global state. Thus in the partially observable setting, each agent in PIC also needs to communicate to get the global state information in the training and the execution. Further details on baselines are given in appendix E.1.

**Neural Network and Parameters.** Recall the intention propagation network is represented by GNN. In our experiment, our graph neural network has $hop = 2$ (2 GNN layers, *i.e.*, $M = 2$) and 1 fully-connected layer at the top. Each layer contains 128 hidden units. Other hyperparameters are listed in appendix H.

## 5.2 COMPARISON TO STATE-OF-THE-ART

In this section, we compare intention propagation (IP) with other baselines. The experiments are evaluated by average episode reward (Lowe et al., 2017). For CityFlow tasks, average reward refers

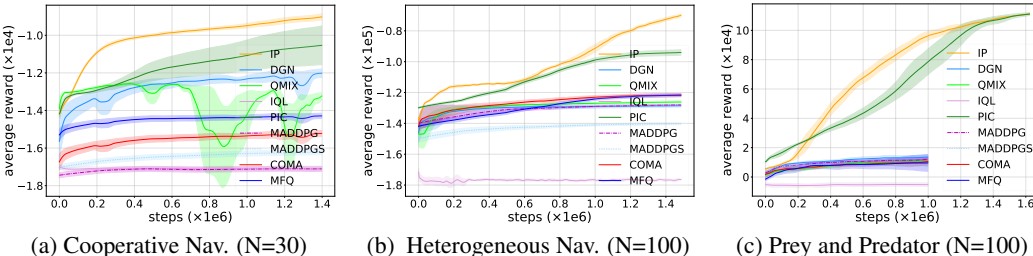

(a) Cooperative Nav. (N=30)  (b) Heterogeneous Nav. (N=100)  (c) Prey and Predator (N=100)

Figure 4: Experimental results on Cooperative Navigation, Heterogeneous Navigation, Prey and Predator. Our intention propagation (IP) beats all the baselines.

to negative average travel time. All experiments are repeated for 5 runs with different random seeds. We report the mean and standard deviation in the curves. We report the results on six experiments and defer all the others to **appendix G** due to the limit of space.

**CityFlow.** We first evaluate our algorithm on traffic control problem. Particularly, we increase the number of intersections (agents) gradually to increase the difficulties of the tasks. Figure 3 presents the performance of different methods on both real-world and synthetic CityFlow data with different number of intersections. On the task of Manhattan City, intention propagation (IP) method, the baseline methods PIC and DGN achieve better reward than the other methods while our method approaches higher reward within fewer steps. On the larger task (N=100), both PIC and DGN have large variance and obtain poor performance. The experiment with N=1225 agents is an extremely challenging task. Our algorithm outperforms all baselines with a wide margin. The runner-up is MADDPG with data shuffling mechanism. Its final performance is around $-4646$ and suffers from large variance. In contrast, the performance of our method is around $-569$ (much higher than the baselines). It's clear that, in both real-world and synthetic cityflow scenarios, the proposed IP method obtains the best performance. We defer further experimental results to appendix G.

**MPE and MAgent.** Figure 4 demonstrates the performance of different methods on other three representative scenario instances: a small task *cooperative navigation* (N=30) and two large-scale tasks *heterogeneous navigation* (N=100) and *prey and predator* (N=100). We run all algorithms long enough (more than 1e6 steps). In all experiments, our algorithm performs best. For *cooperative navigation*, MADDPGS performs better than MADDPG. The potential improvement comes from data-shuffling mechanism, which makes MADDPGS more robust to handle the manually specified order of agents. QMIX performs much better than MADDPG, MADDPGS and IQL. However, its performance is not stable even on small setting (N=30). DGN is better and more stable than QMIX. However, when solving large-scale settings, its performance is much worse than PIC and our intention propagation (IP). Although PIC can solve large-scale tasks, our IP method is still much better. In *prey and predator*, there are two groups of agents: good agents and adversaries. To make a fair comparison of rewards of different methods, we fix good agents' policies and use all the methods to learn the adversaries' policies. Such setting is commonly used in many articles (Lowe et al., 2017; Liu et al., 2019).

**Stability.** Stability is a key criterion to evaluate MARL. In all experiments, our method is quite stable with small variance. For instance, as shown in Figure 3 (b), DGN approaches $-1210 \pm 419$ on the CityFlow scenario with N=100 intersections while our method approaches $-465 \pm 20$ after $1.6 \times 10^6$ steps (much better and stable). The reason is that to make the joint decision, the agent in our algorithm can adjust its own policy *properly* by considering other agents' plans.

**Ablation Study:** We conduct a set of ablation studies related to the effect of joint policy, graph, hop size, number of neighbors and the assumption of the reward function. Particularly, we find the joint policy is essential for the good performance. In Cityflow, the performance of traffic graph (2-d grid induced by the roadmap) is better than the fully connected graph. In MPE and MAgent, We define the adjacent matrix based on the $k$ nearest neighbors and pick $k = 8$ in large scale problem and $k = 4$ in small scale problem. In all of our experiment, we choose the 2-hop GNN. Because of the limitation of space, we just summarize our conclusion here and place the details in appendix F.

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

## A    ORGANIZATION OF THE APPENDIX

In appendix **B**, we give the details on the intention propagation network and parameterization of the GNN. We explain intention propgation from the view of the MARL. At last, we extend the intention propagation to other approximations which converges to other solutions of the variational inference. Notice such extension on the algorithm can also be easily parameterized by neural networks.

In Appendix **C**, we give the details of the algorithm deferred from the main paper. Appendix **D** summarizes the configuration of the experiment and MARL environment. Appendix **E** gives more details on baselines and the hyperparameters of GNN used in our model. Appendix **F** conducts the ablation study deferred from the main paper. Appendix **G** and **H** give more experimental results and hyperparameters used in the algorithms. At appendix I, we derive the algorithm and prove the proposition 1.

## B    INTENTION PROPAGATION NETWORK

### B.1    DETAILS ON THE INTENTION PROPAGATION NETWORK

In this section, we give the details on the intention propagation network deferred from the main paper. We first illustrate the message passing of the intention propagation derived in section 4.1. Then we give a details on how to construct graph neural network.

**Message passing and explanation from the view of MARL:** $\tilde{\mu}_i$ is the embedding of *policy* of agent $i$, which represents the intention of the agent $i$. At 0 iteration, every agent makes independent decision. The policy of agent $i$ is mapped into its embedding $\tilde{\mu}_i^0$. We call it the intention of agent $i$ at iteration 0. Then agent $i$ sends its plan to its neighbors . In Figure 5, $\tilde{\mu}_i^m$ is the $d$ dimensional ($d = 3$ in this figure) embedding of $q_i$ at $m-$th iteration of intention propagation. We draw the update of $\tilde{\mu}_1^{(m)}$ as example. Agent 1 receives the embedding (intention) $\tilde{\mu}_2^{m-1}, \tilde{\mu}_5^{m-1}, \tilde{\mu}_6^{m-1}$ from its neighbors, and then updates the its own embedding with operator $\tilde{\mathcal{T}}$. After $M$ iterations, we obtain $\tilde{\mu}_1^M$ and output the policy distribution $q_1$ using equation 4. Similar procedure holds for other agents. At each RL step $t$, we do this procedure (with M iterations) once to generate joint policy. $M$ in general is small, e.g., $M = 2$ or 3. Thus it is efficient.

**Parameterization on GNN:** We then illustrate the parameterization of graph neural network in Figure 6. If the action space is discrete, the output $q_i(a_i|s)$ is a softmax function. When it is continuous, we can output a Gaussian distribution (mean and variance) with the reparametrization trick (Kingma & Welling, 2019). Here, we draw 2-hop (layer) GNN to parameterize it in discrete action intention propagation. In Figure 6 (b), each agent observe its own state $s_i$. After a MLP and softmax layer (we do not sample here, and just use the output probabilities of the actions), we get a embedding $\tilde{\mu}_i^0$, which is the initial distribution of the policy. In the following, we use agent 1 as an example. To ease the exposition, we assume Agent 1 just has one neighbor, agent 2. Agent 1 receives the embedding $\tilde{\mu}_2^0$ of its neighbor. After a GNN layer to combine the information, e.g, $Relu[W_1(s_1 + s_2) + W_2(\tilde{\mu}_1^0 + \tilde{\mu}_2^0)]$, we obtain new embedding $\tilde{\mu}_1^1$ of agent 1. Notice we also do

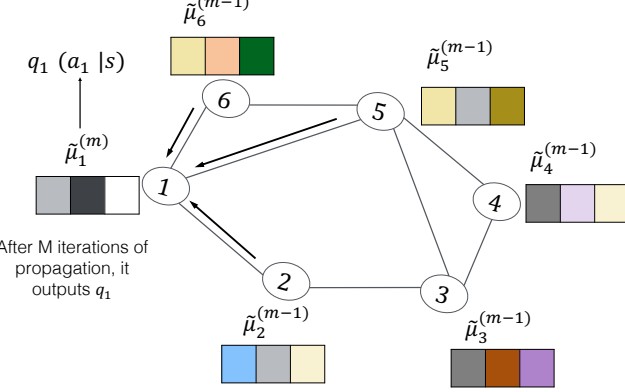

Figure 5: illustrate the message passing in intention propagation network $\Lambda_\theta(\mathbf{a}|s)$.

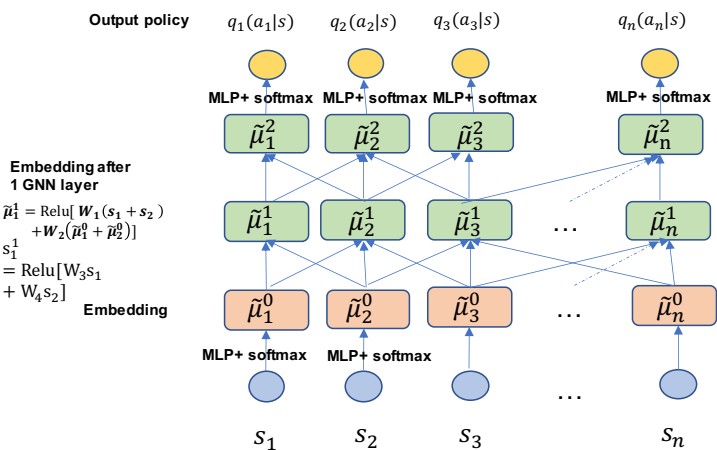

Figure 6: Details of the graph neural network

message passing on state, since in practice the global state is not available. In the second layer, we do similar things. Agent 1 receives the embedding information of $\tilde{\mu}_2^1$ from its neighbors and get a new embedding $\tilde{\mu}_1^2$. Then this embedding passes a MLP+softmax layer and output probability of action, i.e. $q_1(a_1|s)$.

## B.2 EXTENSION TO OTHER VARIATIONAL INFERENCE METHODS AND NEURAL NETWORKS

In this section, we show how to approximate the joint policy with the Loopy Belief Propagation in the variational inference (Yedidia et al., 2001). This will lead to a new form of neural networks beyond vanilla GNN that we illustrate above.

The objective function in Loop Belief Propagation is the Beth Free energy (Yedidia et al., 2001). Different from the mean-field approximation, it introduces another variational variable $q_{ij}$, which brings more flexibility on the approximation. The following is objective function in our case.

$$\min_{q_i, q_{ij} \in \mathcal{E}} - \sum_i (|\mathcal{N}_i| - 1) \int q_i(a_i|s) \log \frac{q_i(a_i|s)}{\psi_i(s, a_i)} da_i$$

$$+ \sum_{ij} \int q_{ij}(a_i, a_j|s) \log \frac{q_{ij}(a_i, a_j|s)}{\psi_{ij}(s, a_i, a_j)\psi_i(s, a_i)\psi_j(s, a_j)} da_i da_j. \quad (6)$$

$$s.t. \int q_{ij}(a_i, a_j|s) da_j = q_i(a_j|s), \int q_{ij}(a_i, a_j|s) da_i = q_j(a_j|s)$$

Solve above problem, we have the fixed point algorithm

$$m_{ij}(a_j|s) \leftarrow \int \prod_{k \in \mathcal{N}_i \setminus j} m_{ki}(a_i|s)\psi_i(s, a_i)\psi_{ij}(s, a_i, a_j) da_i,$$

$$q_i(a_i|s) \leftarrow \psi_i(s, a_i) \prod_{j \in \mathcal{N}_i} m_{ji}(a_i|s).$$

Similar to the mean-field approximation case, we have

$$m_{ij}(a_j|s) = f(a_j, s, \{m_{ki}\}_{k \in \mathcal{N}_i \setminus j}),$$

$$q_i(a_i|s) = g(a_i, s, \{m_{ki}\}_{k \in \mathcal{N}_i}),$$

It says the message $m_{ij}$ and marginals $q_i$ are functionals of messages from neighbors.

Denote the embedding $\tilde{\nu}_{ij} = \int \psi_j(s, a_j)m_{ij}(a_j|s)da_j$ and $\tilde{\mu}_i = \int \psi_i(s, a_i)q_i(a_i|s)da_i$, we have

$\tilde{\nu}_{ij} = \tilde{\mathcal{T}} \circ \left(s, \{\tilde{\nu}_{ki}\}_{k \in \mathcal{N}_i \setminus j}\right), \tilde{\mu}_i = \tilde{T} \circ \left(s, \{\tilde{\nu}_{ki}\}_{k \in \mathcal{N}_i}\right).$

Again, we can parameterize above equation by (graph) neural network $\tilde{\nu}_{ij} = \sigma\big(W_1 s + W_2 \sum_{k \in \mathcal{N}_i \setminus j} \tilde{\nu}_{ki}\big), \tilde{\mu}_i = \sigma\big(W_3 s + W_4 \sum_{k \in \mathcal{N}_i} \tilde{\nu}_{ki}\big)$.

Following similar way, we can derive different intention propagation algorithms by changing different objective function which corresponds to e.g., double-loop belief propagation(Yuille, 2002), tree-reweighted belief propagation (Wainwright et al., 2003) and many others.

## C  ALGORITHM

We present some remarks of the algorithm Intention Propagation (algorithm 1) deferred from the main paper.

Remark: To calculate the loss function $J(\eta_i)$, each agent need to sample the global state and $(a_i, a_{\mathcal{N}_i})$. Thus we first sample a global state from the replay buffer and then sample all action $\mathbf{a}$ once using the intention propagation network.

## D  FURTHER DETAILS ABOUT ENVIRONMENTS AND EXPERIMETAL SETTING

Table 1 summarizes the setting of the tasks in our experiment.

Table 1: Tasks. We evaluate MARL algorithms on more than 10 different tasks from three different environments.

| Env | Scenarios | #agents (N) |
|---|---|---|
| CityFlow | Realworld:Hang Zhou | N=16 |
| | Realworld:Manhattan | N=96 |
| | Synthetic Map | N=49, 100, 225, 1225 |
| MPE | Cooperative Nav. | N=15, 30, 200 |
| | Heterogeneous Nav. | N=100 |
| | Cooperative Push | N=100 |
| | Prey and Predator | N=100 |
| MAgent | Jungle | N=20, F=12 |

### D.1  CITYFLOW

CityFlow (Tang et al., 2019) is an open-source MARL environment for large-scale city traffic signal control [1]. After the traffic road map and flow data being fed into the simulators, each vehicle moves from its origin location to the destination. The traffic data contains bidirectional and dynamic flows with turning traffic. We evaluate different methods on both real-world and synthetic traffic data. For real-world data, we select traffic flow data from Gudang sub-district, Hangzhou, China and Manhattan, USA [2]. For synthetic data, we simulate several different road networks: $7 \times 7$ grid network ($N = 49$) and large-scale grid networks with $N = 10 \times 10 = 100$, $15 \times 15 = 225$, $35 \times 35 = 1225$. Each traffic light at the intersection is the agent. In the real-world setting (Hang Zhou, Manhattan), the graph is a 2-d grid induced by the roadmap. Particularly, the roads are edges which connect the node (agent) of the graph. For the synthetic data, the map is a $n * n$ 2-d grid (Something like Figure 7), where edges represents road, node is the traffic light. We present the experimental results deferred from the main paper in Figure 10.

### D.2  MPE

In MPE (Mordatch & Abbeel, 2017) [3], the observation of each agent contains relative location and velocity of neighboring agents and landmarks. The number of visible neighbors in an agent's observation is equal to or less than 10. In some scenarios, the observation may contain relative location and velocity of neighboring agents and landmarks.

---

[1]https://github.com/cityflow-project/CityFlow

[2]We download the maps from https://github.com/traffic-signal-control/sample-code.

[3]To make the environment more computation-efficient, Liu et al. (2019) provided an improved version of MPE. The code are released in https://github.com/IouJenLiu/PIC.

We consider four scenarios in MPE. (1) *cooperative navigation*: $N$ agents work together and move to cover $L$ landmarks. If these agents get closer to landmarks, they will obtain a larger reward. In this scenario, the agent observes its location and velocity, and the relative location of the nearest 5 landmarks and $N$ agents. The observation dimension is 26. (2) *prey and predator*: $N$ slower cooperating agents must chase the faster adversaries around a randomly generated environment with $L$ large landmarks. Note that, the landmarks impede the way of all agents and adversaries. This property makes the scenario much more challenging. In this scenario, the agent observes its location and velocity, and the relative location of the nearest 5 landmarks and 5 preys. The observation dimension is 34. (3) *cooperative push*: $N$ cooperating agents are rewarded to push a large ball to a landmark. In this scenario, each agent can observe 10 nearest agents and 5 nearest landmarks. The observation dimension is 28. (4) *heterogeneous navigation*: this scenario is similar with cooperative navigation except dividing $N$ agents into $\frac{N}{2}$ big and slow agents and $\frac{N}{2}$ small and fast agents. If small agents collide with big agents, they will obtain a large negative reward. In this scenario, each agent can observe 10 nearest agents and 5 nearest landmarks. The observation dimension is 26.

Further details about this environment can be found at `https://github.com/IouJenLiu/PIC`.

### D.3 MAGENT

MAgent (Zheng et al., 2018) is a grid-world platform and serves another popular environment platform for evaluating MARL algorithms. Jiang et al. (2020) tested their method on two scenarios: *jungle* and *battle*. In *jungle*, there are $N$ agents and $F$ foods. The agents are rewarded by positive reward if they eat food, but gets higher reward if they attack other agents. This is an interesting scenario, which is called by *moral dilemma*. In *battle*, $N$ agents learn to fight against several enemies, which is very similar with the *prey and predator* scenario in MPE. In our experiment, we evaluate our methods on *jungle*.

In our experiment, the size for the grid-world environment is $30 \times 30$. Each agent refers to one grid and can observe $11 \times 11$ grids centered at the agent and its own coordinates. The actions includes moving and attacking along the coordinates. Further details about this environment can be found at `https://github.com/geek-ai/MAgent` and `https://github.com/PKU-AI-Edge/DGN`.

## E FURTHER DETAILS ON SETTINGS

### E.1 DESCRIPTION OF OUR BASELINES

We compare our method with multi-agent deep deterministic policy gradient (MADDPG) (Lowe et al., 2017), a strong actor-critic algorithm based on the framework of centralized training with decentralized execution; QMIX (Rashid et al., 2018), a q-learning based monotonic value function factorisation algorithm; permutation invariant critic (PIC) (Liu et al., 2019), a leading algorithm on MPE yielding identical output irrespective of the agent permutation; graph convolutional reinforcement learning (DGN) (Jiang et al., 2020), a deep q-learning algorithm based on deep convolutional graph neural network with multi-head attention, which is a leading algorithm on MAgent; independent Q-learning (IQL) (Tan, 1993), decomposing a multi-agent problem into a collection of simultaneous single-agent problems that share the same environment, which usually serves as a surprisingly strong benchmark in the mixed and competitive games (Tampuu et al., 2017). In homogeneous settings, the input to the centralized critic in MADDPG is the concatenation of all agent's observations and actions along the specified agent order, which doesn't hold the property of *permutation invariance*. We follow the similar setting in (Liu et al., 2019) and shuffle the agents' observations and actions in training batch [4]. In COMA (Foerster et al., 2018), it directly assume the poilcy is factorized. It calculates the counterfactual baseline to address the credit assignment problem in MARL. In our experiment, since we can observe each reward function, each agent can directly approximate the Q function without counterfactual baseline. MFQ derives the algorithm from the view of mean-field game(Yang et al., 2018). Notice the aim of mean-field game is to find the Nash equilibrium rather

---

[4]This operation doesn't change the state of the actions.

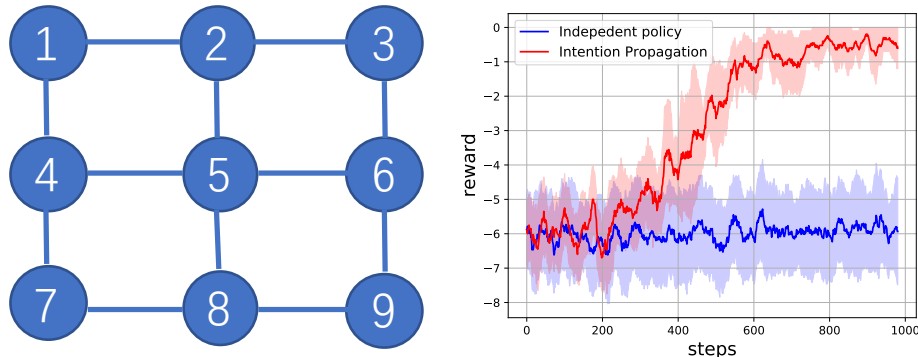

Figure 7: (a) a toy task on 2d-grid. (b) The performance of independent policy and intention propagation.

than maxmization of the total reward of the group. Further more, it needs the assumption that agents are identical.

### E.2 NEURAL NETWORKS ARCHITECTURE

To learn feature from structural graph build by the space distance for different agents, we design our graph neural network based on the idea of a strong graph embedding tool *structure2vec* (Dai et al., 2016), which is an effective and scalable approach for structured data representation through embedding latent variable models into feature spaces. Structure2vec extracts features by performing a sequence of function mappings in a way similar to graphical model inference procedures, such as mean field and belief propagation. After using $M$ graph neural network layers, each node can receive the information from $M$-hops neighbors by message passing. Recently, attention mechanism empirically leads to more powerful representation on graph data (Veličković et al., 2017; Jiang et al., 2020). We employ this idea into our graph neural network. In some settings, such as *heterogeneous navigation* scenario from MPE, the observations of different group of agents are heterogeneous. To handle this issue, we use different nonlinear functions to extract the features from heterogeneous observations and map the observations into a latent layer, then use the same graph neural networks to learn the policy for all types of agents. In our experiment, our graph neural network has $M = 2$ layers and 1 fully-connected layer at the top. Each layer contains 128 hidden units.

## F ABLATION STUDIES

### F.1 INDEPENDENT POLICY VS INTENTION PROPAGATION.

We first give a toy example where the independent policy (without communication) fails. To implement such algorithm, we just replace the intention propagation network by a independent policy network and remain other parts the same. Think about a $3 \times 3$ 2d-grid in Figure 7 where the global state (can be observed by all agents) is a constant scalar (thus no information). Each agent chooses an action $a_i = 0$ or 1. The aim is to maximize a reward $-(a_1 - a_2)^2 - (a_1 - a_4)^2 - (a_2 - a_3)^2 - ... - (a_8 - a_9)^2$, (i.e., summation of the reward function on edges). Obviously the optimal value is 0. The optimal policy for agents is $a_1 = a_2 =, ..., a_9 = 0$ or $a_1 = a_2 =, ..., a_9 = 1$. However independent policy fails, since each agents does not know how its allies pick the action. Thus the learned policy is random. We show the result of this toy example in Figure 7, where intention propagation learns optimal policy.

### F.2 GRAPH TYPES, NUMBER OF NEIGHBORS, AND HOP SIZE

We conduct a set of ablation studies related to graph types, the number of neighbors, and hop size. Figure 8(a) and Figure 8(b) demonstrate the performance of our method on traffic graph and fully-connected graph on the scenarios (N=49 and N=100) of CityFlow. In the experiment, each agent can only get the information from its neighbors through message passing (state embedding and the policy embedding). The result makes sense, since the traffic graph represents the structure of the

map. Although the agent in the fully connected graph would obtain global information, it may introduce irrelevant information from agents far away.

Figure 8(c) and Figure 8(d) demonstrate the performance under different number of neighbors and hop size on *cooperative navigation* (N=30) respectively. The algorithm with neighbors=8 has the best performance. Again the the fully connected graph (neighbors=30) may introduce the irrelevant information of the agents far away. Thus its performance is worse than the algorithm with graph constructed by the K-nearest neighbor. In addition the fully connected graph introduces more computations in the training. In Figure 8(d), we increase the hop-size from 1 to 3. The performance of IP with hop=2 is much better than that with hop=1. While IP with hop=3 is just slightly better than that with hop=2. It means graph neural network with hop size =2 has aggregated enough information.

In Figure 8(e), we test the importance of the k-nearest neighbor structure. IP(neighbors=3)+random means that we pick 3 agents uniformly at random as the neighbors. Obviously, IP with K-nearest neighbors outperforms the IP with random graph a lot. In Figure 8(f), we update adjacency matrix every 1, 5, 10 steps. IP(neighbors=8) denotes that we update the adjacency matrix every step, while IP(neighbors=8)+reset(5) and IP(neighbors=8)+reset(10) denote that we update adjacency matrix every 5 and 10 steps respectively. Obviously, IP(neighbors=8) has the best result. IP(neighbors=8)+reset(5) is better than IP(neighbors=8)+reset(10). The result makes sense, since the adjacency matrix is more accurate if the update interval is smaller.

### F.3 ASSUMPTION VIOLATION

The aforementioned experimental evaluations are based on the mild assumption: the actions of agents that are far away would not affect the learner because of their physical distance. It would be interesting to see the performance where the assumption is violated. As such, we modify the reward in the experiment of cooperative navigation. In particular, the reward is defined by $r = r1 + r2$, where $r1$ encourages the agents to cover (get close to) landmarks and $r2$ is the log function of the distances between agents (farther agents have larger impact). To make a violation, we let $r2$ dominate the reward. We conduct the experiments with $hop = 1, 2, 3$. Figure 9 shows that the rewards obtained by our methods are $4115 \pm 21$, $4564 \pm 22$, and $4586 \pm 25$ respectively. It's expected in this scenario, since we should use large hop to collect information from the far-away agents.

### G FURTHER EXPERIMENTAL RESULTS

For most of the experiments, we run them long enough with 1 million to 1.5 million steps and stop (even in some cases our algorithm does not converge to the asymptotic result), since every experment in MARL may cost several days. We present the results on Cityflow in Figure 10. Figure 11 provides the experimental results on the cooperative navigation instances with $N = 15$, $N = 30$ and $N = 200$ agents. Note that, the instance with $N = 200$ is a large-scale and challenging multi-agents reinforcement learning setting (Chen et al., 2018; Liu et al., 2019), which typically needs several days to run millions of steps. It's clear that IQL, MADDPG, MADDPG perform well in the small setting (N=15), however, they failed in large-scale instances ($N = 30$ and $N = 200$). In the instance with $N = 30$, MADDPGS performs better than MADDPG. The potential reason is that with the help of shuffling, MADDPGS is more robust to handle the manually specified order of agents. Although QMIX performs well in the instance of $N = 15$ and $N = 30$, it has large variances in both settings. DGN using graph convolutional network can hold the property of permutation invariance, it obtains much better performance than QMIX on these two settings. However, it also fails to solve the large-scale settings with $N = 200$ agents. Empirically, after $1.5 \times 10^6$ steps, PIC obtains a large reward ($-425085 \pm 31259$) on this large-scale setting. Despite all these, the proposed intention propagation (IP) approaches $-329229 \pm 14730$ and is much better than PIC. Furthermore, Figure 11 shows the results of different methods on (d) jungle (N=20, F=12) and (e) *prey and predator* (N=100). The experimental results shows our method can beats all baselines on these two tasks. On the scenario of *cooperative push* (N=100) as shown in Figure 11(f), it's clear that DGN, QMIX, IQL, MADDPG and MADDPGS all fail to converge to good rewards after $1.5 \times 10^6$ environmental steps. In contrast, PIC and the proposed IP method obtain much better rewards than these baselines. Limited by the computational resources, we only show the long-term performance of the best two methods. Figure 11(f) shows that IP is slightly better than PIC in this setting.

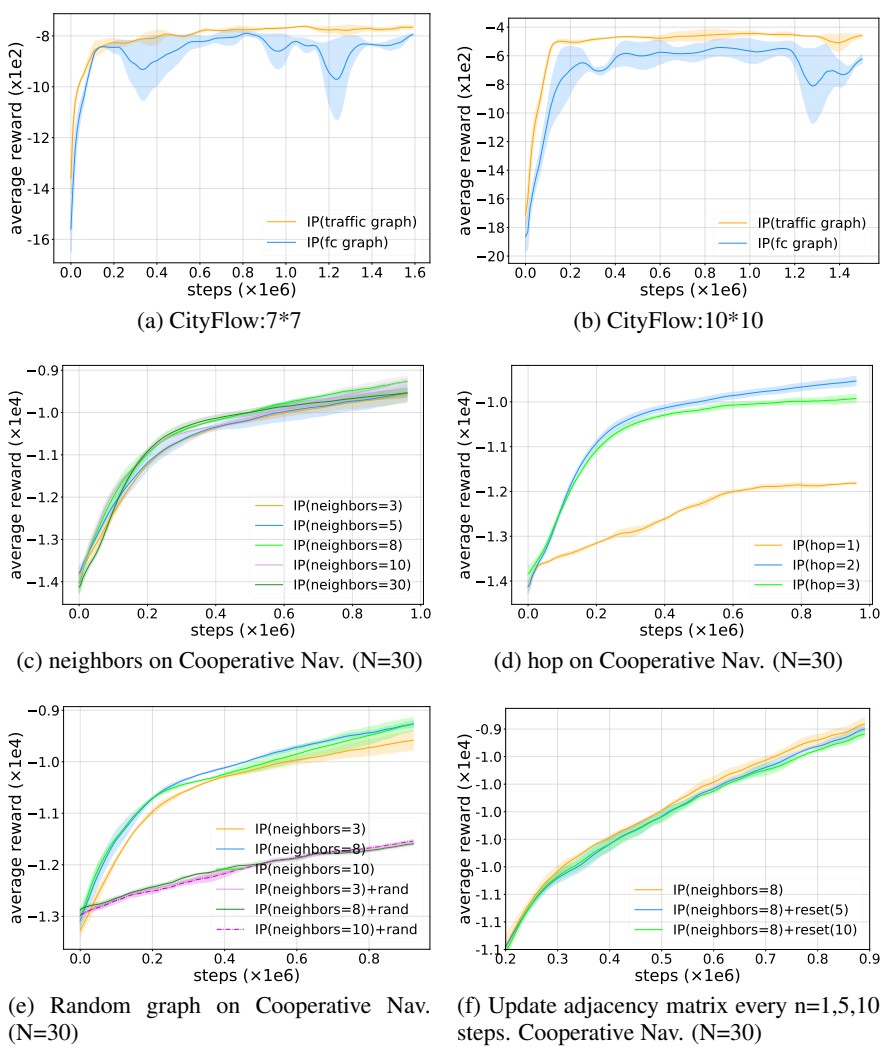

Figure 8: Performance of the proposed method based on different ablation settings. (a) Traffic graph and fully connected (fc) graph on CityFlow (N=49). (b) Traffic graph and fully connected (fc) graph on CityFlow (N=100). (c) Cooperative Nav. (N=30): Different number of neighbors. (d) Cooperative Nav. (N=30): Different hop size graph neural networks. (e) Cooperative Nav. (N=30): Construct random graph vs $k$-nearest-neighbor graph ($k = 3, 8, 10$). (f) Cooperative Nav. (N=30): Update $8$-nearest-neighbor graph every n environment steps (5 and 10 respectively.).

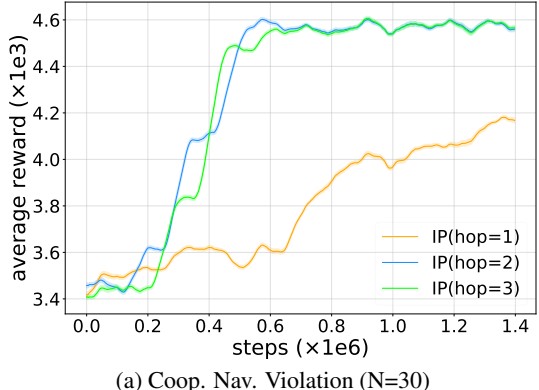

(a) Coop. Nav. Violation (N=30)

Figure 9: Further experimental results. Cooperative navigation (N=30) with assumption violation.

### G.1 POLICY INTERPRETATION

Explicitly analyzing the policy learned by deep multi-agent reinforcement learning algorithm is a challenging task, especially for the large-scale problem. We follow the similar ideas from (Zheng et al., 2019) and analyze the learned policy on CityFlow in the following way: We select the same period of environmental steps within [210000, 1600000] and group these steps into 69 intervals (each interval contains about 20000 steps). We compute the ratio of vehicle volume on each movement and the sampled action volume from the learned policy (each movement can be assigned to one action according to the internal function in CityFlow). We define the ratio of vehicle volume over all movements as the *vehicle volume distribution* and define the ratio of the sampled action volume from the learned policy over all movements as the *sampled action distribution*. It's expected that a good MARL algorithm will hold the property: these two distributions will very similar over a period of time. Figure 12 reports their KL divergence by intervals. It's clear that the proposed intention propagation method (IP) obtains the lowest KL divergence (much better than the state-of-the-art baselines). Because KL divergence is not symmetrical metric, we also calculate their Euclidean distances. Specifically, the distance of our method is $0.0271$ while DGN is $0.0938$ and $PIC$ is $0.0933$.

## H HYPERPARAMETERS

The parameter on the environment. For the max episode length, we follow the similar settings like that in the baselines (Lowe et al., 2017) . Particularly, we set 25 for MPE and set 100 for CityFlow. For MAgent, we find that setting the max episode length by 25 is better than 100. All the methods share the same setting.

We list the range of hyperparameter that we tune in all baselines and intention propagation. $\gamma$ : $\{0.95, 0.98, 0.99, 0.999\}$, learning rate : $\{1, 5, 10, 100\} \times$1e-4. activation function: $\{relu, gelu, tanh\}$, batch size:$\{128, 256, 512, 1024\}$, gradient steps: $\{1, 2, 4, 8\}$. Number of hidden units in MLP: $\{32, 64, 128, 256, 512\}$, number of layers in MLP:$\{1, 2, 3\}$ in all experiment. In Qmix, GRU hidden unites are $\{64, 128\}$. A fully connected layer is before and after GRU. Hypernetwork and mixing network are both single layer network(64 hidden units with Relu activation from the Qmix paper). The parameter of intention propagation is reported in Table.2.

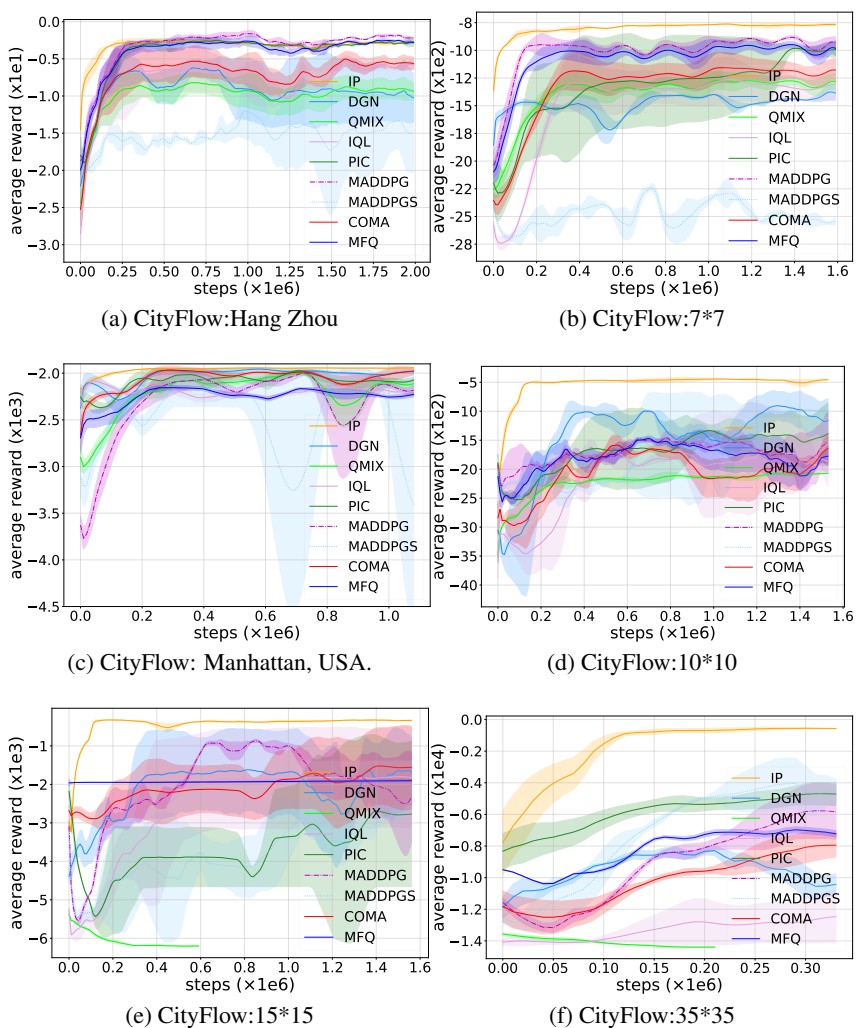

Figure 10: Performance of different methods on traffic lights control scenarios in CityFlow environment: (a) N=16 (4 × 4 grid), Gudang sub-district, Hangzhou, China. (b) N=49 (7 × 7 grid), (c) N=96 (irregular grid map), Manhattan, USA. (d) N=100 (10 × 10 grid), (e) N=225 (15 × 15 grid), (f) N=1225 (35 × 35 grid). The horizontal axis is time steps (interaction with the environment). The vertical axis is average episode reward, which refers to negative average travel time. Higher rewards are better. The proposed intention propagation (IP) obtains much better performance than all the baselines on large-scale tasks .

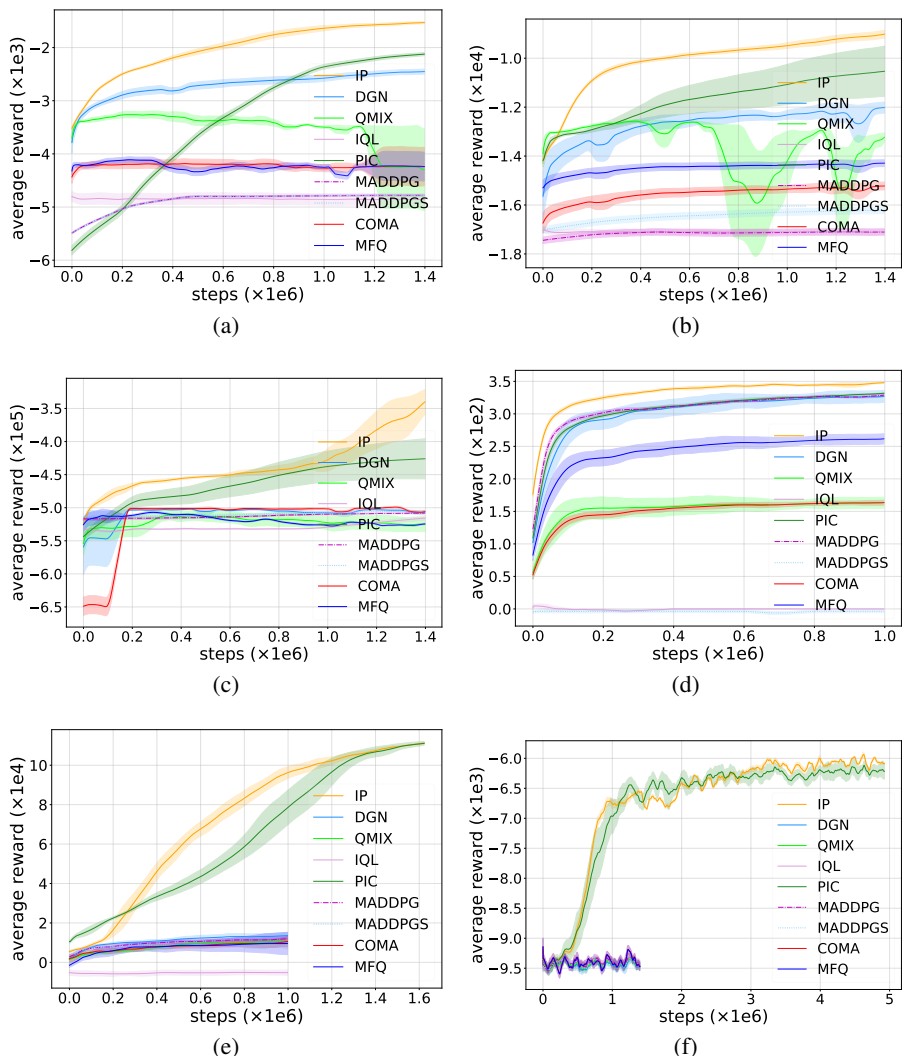

(a)

(b)

(c)

(d)

(e)

(f)

Figure 11: Comparison on instances with different number of agents: *cooperative navigation* (a) N=15, (b) N=30 and (c) N=200 respectively. (d) *jungle* (N=20, F=12), (e) *prey and predator* (N=100), and (f) *cooperative push* (N=100). The horizontal axis is environmental steps (number of interactions with the enviroment). The vertical axis is average episode reward. The larger average reward indicates better result. The proposed intention propagation (IP) beats all the baselines on different scale of instances.

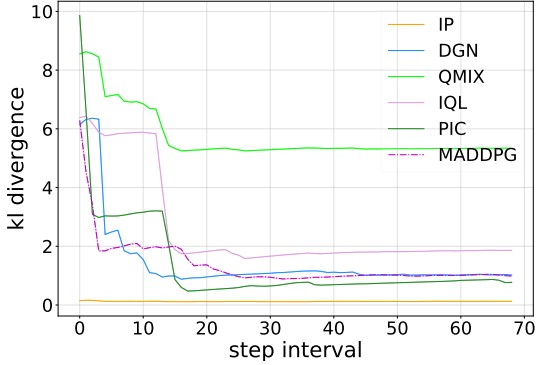

Figure 12: Policy interpreation on CityFlow task (N=49)

Table 2: Hyperparameters

| Parameter | Value |
|---|---|
| optimizer | Adam |
| learning rate of all networks | 0.01 |
| discount of reward | 0.95 |
| replay buffer size | $10^6$ |
| max episode length in MAgent | 25 |
| max episode length in MPE, CityFlow | 100 |
| number of hidden units per layer | 128 |
| number of samples per minibatch | 1024 |
| nonlinearity | ReLU |
| target smoothing coefficient ($\tau$) | 0.01 |
| target update interval | 1 |
| gradient steps | 8 |
| regularizer factor($\alpha$) | 0.2 |

# I DERIVATION

## I.1 PROOF OF PROPOSITION 1

We prove the result by induction using the backward view.

To see that, plug $r(s^t, \mathbf{a}^t) = \sum_{i=1}^{N} r_i(s^t, a_i^t, a_{\mathcal{N}_i}^t)$ into the distribution of the optimal policy defined in section 3.

$$p(\tau) = [p(s^0) \prod_{t=0}^{T} p(s^{t+1}|s^t, \mathbf{a}^t)] \exp \sum_{t=0}^{T} \sum_{i=1}^{N} r_i(s^t, a_i^t, a_{\mathcal{N}_i}^t)$$

Recall the goal is to find the best approximation of $\pi(\mathbf{a}^t|s^t)$ such that the trajectory distribution $\hat{p}(\tau)$ induced by this policy can match the optimal trajectory probability $p(\tau)$. Thus we minimize the KL divergence between them $\min_\pi D_{KL}(\hat{p}(\tau)\|p(\tau))$, where $\hat{p}(\tau) = p(s^0) \prod_{t=0}^{T} p(s^{t+1}|s^t, \mathbf{a}^t)\pi(\mathbf{a}^t|s^t)$. We can do optimization w.r.t. $\pi(a^t|s^t)$ as that in (Levine, 2018) and obtain a backward algorithm on the policy $\pi^*(\mathbf{a}^t|s^t)$ (See equation 13 in I.2.)

$$\pi^*(\mathbf{a}^t|s^t) = \frac{1}{Z} \exp\left(\mathbb{E}_{p(s^{t+1:T}, \mathbf{a}^{t+1:T}|s^t, \mathbf{a}^t)}[\sum_{t'=t}^{T} \sum_{i=1}^{N} r_i(s^{t'}, a_i^{t'}, a_{\mathcal{N}_i}^{t'}) - \sum_{t'=t+1}^{T} \log \pi(\mathbf{a}^{t'}|s^{t'})]\right). \quad (7)$$

Using the result equation 7, when $t = T$, the optimal policy is

$$\pi^*(\mathbf{a}^T|s^T) = \frac{1}{Z} \exp(\sum_{i=1}^{N} r_i(s^T, a_i^T, a_{\mathcal{N}_i}^T)).$$

Obviously, it satisfies the form $\pi^*(\mathbf{a}^T|s^T) = \frac{1}{Z} \exp(\sum_{i=1}^{N} \psi_i(s^T, a_i^T, a_{\mathcal{N}_i}^T))$.

Now suppose from step $t + 1$ to $T$, we have

$$\pi^*(\mathbf{a}^{t'}|s^{t'}) = \frac{1}{Z} \exp(\sum_{i=1}^{N} \psi_i(s^{t'}, a_i^{t'}, a_{\mathcal{N}_i}^{t'})) \quad (8)$$

for $t' = t + 1, ..., T$.

Recall that we have the result

$$\pi^*(\mathbf{a}^t|s^t) = \frac{1}{Z} \exp\left(\mathbb{E}_{p(s^{t+1:T}, \mathbf{a}^{t+1:T}|s^t, \mathbf{a}^t)}[\sum_{t'=t}^{T} \sum_{i=1}^{N} r_i(s^{t'}, a_i^{t'}, a_{\mathcal{N}_i}^{t'}) - \sum_{t'=t+1}^{T} \log \pi^*(\mathbf{a}^{t'}|s^{t'})]\right).$$

$$(9)$$

Now plug equation 8 into equation 9, we have

$$\pi^*(\mathbf{a^t}|s^t) = \frac{1}{Z} \exp\left(\mathbb{E}_{p(s^{t+1:T}, \mathbf{a}^{t+1:T}|s^t, \mathbf{a^t})}[\sum_{t'=t}^{T}\sum_{i=1}^{N} r_i(s^{t'}, a_i^{t'}, a_{\mathcal{N}_i}^{t'}) - \sum_{t'=t+1}^{T}\sum_{i=1}^{N} \psi_i(s_i^{t'}, a_i^{t'}, a_{\mathcal{N}_i}^{t'}) + C]\right),$$
$$(10)$$

where $C$ is some constant related to the normalization term. Thus, we redefine a new term

$$\tilde{\psi}_i(s^t, a^t, a_{\mathcal{N}_i}^t) = \mathbb{E}_{p(s^{t+1:T}, a^{t+1:T}|s^t, a^t)}\left[\sum_{t=t'}^{T}\left(r_i(s^{t'}, a_i^{t'}, a_{\mathcal{N}_i}^{t'}) - \sum_{t'=t+1}^{T}\psi_i(s^{t'}, a^{t'}, a_{\mathcal{N}_i}^{t'})\right)\right]. \quad (11)$$

Then obviously $\pi^*(\mathbf{a}^t|s^t)$ satisfies the form what we need by absorbing the constant $C$ into the normalization term . Thus we have the result.

## I.2 DERIVATION OF THE ALGORITHM

We start the derivation with minimization of the KL divergence $KL(\hat{p}(\tau)||p(\tau))$, where $p(\tau) = [p(s^0)\prod_{t=0}^{T}p(s^{t+1}|s^t, \mathbf{a^t})]\exp\left(\sum_{t=0}^{T}\sum_{i=1}^{N}r_i(s^t, a_i^t, a_{\mathcal{N}_i}^t)\right)$, $\hat{p}(\tau) = p(s^0)\prod_{t=0}^{T}p(s^{t+1}|s^t, \mathbf{a^t})\pi(\mathbf{a^t}|s^t)$.

$$KL(\hat{p}(\tau)||p(\tau)) = \mathbb{E}_{\tau \sim \hat{p}(\tau)}\sum_{t=0}^{T}\left(\sum_{i=1}^{N}r_i(s^t, a_i^t, a_{\mathcal{N}_i}^i) - \log\pi(\mathbf{a^t}|s^t)\right)$$
$$= \sum_{\tau}[p(s^0)\prod_{t=0}^{T}p(s^{t+1}|s^t, \mathbf{a^t})\pi(\mathbf{a^t}|s^t)]\sum_{t=0}^{T}\left(\sum_{i=1}^{N}r_i(s^t, a_i^t, a_{\mathcal{N}_i}^t) - \log\pi(\mathbf{a^t}|s^t)\right). \quad (12)$$

Now we optimize KL divergence w.r.t $\pi(\cdot|s^t)$. Considering the constraint $\sum_j \pi(j|s^t) = 1$, we introduce a Lagrangian multiplier $\lambda(\sum_{j=1}^{|\mathcal{A}|}\pi(j|s^t) - 1)$ (Rigorously speaking, we need to consider another constraint that each element of $\pi$ is larger than 0, but later we will see the optimal value satisfies this constraint automatically). Now we take gradient of $KL(\hat{p}(\tau)||p(\tau)) + \lambda(\sum_{j=1}^{|\mathcal{A}|}\pi(j|s^t) - 1)$ w.r.t $\pi(\cdot|s)$, set it to zero, and obtain

$$\log\pi^*(\mathbf{a^t}|s^t) = \mathbb{E}_{p(s^{t+1:T}, \mathbf{a}^{t+1:T}|s^t, \mathbf{a^t})}[\sum_{t'=t}^{T}\sum_{i=1}^{N}r_i(s^{t'}, a_i^{t'}, a_{\mathcal{N}_i}^{t'}) - \sum_{t'=t+1}^{T}\log\pi(\mathbf{a}^{t'}|s^{t'})] - 1 + \lambda.$$

Therefore

$$\pi^*(\mathbf{a^t}|s^t) \propto \exp\left(\mathbb{E}_{p(s^{t+1:T}, \mathbf{a}^{t+1:T}|s^t, \mathbf{a^t})}[\sum_{t'=t}^{T}\sum_{i=1}^{N}r_i(s^{t'}, a_i^{t'}, a_{\mathcal{N}_i}^{t'}) - \sum_{t'=t+1}^{T}\log\pi(\mathbf{a}^{t'}|s^{t'})]\right).$$

Since we know $\sum_j \pi(j|s^t) = 1$, thus we have

$$\pi^*(\mathbf{a^t}|s^t) = \frac{1}{Z}\exp\left(\mathbb{E}_{p(s^{t+1:T}, \mathbf{a}^{t+1:T}|s^t, \mathbf{a^t})}[\sum_{t'=t}^{T}\sum_{i=1}^{N}r_i(s^{t'}, a_i^{t'}, a_{\mathcal{N}_i}^{t'}) - \sum_{t'=t+1}^{T}\log\pi(\mathbf{a}^{t'}|s^{t'})]\right). \quad (13)$$

For convenience, we define the soft $V$ function and $Q$ function as that in (Levine, 2018), and will show how to decompose them into $V_i$ and $Q_i$ later.

$$V(s^{t+1}) := \mathbb{E}\big[ \sum_{t'=t+1}^{T} \sum_{i=1}^{N} r_i(s^{t'}, a_i^{t'}, a_{\mathcal{N}_i}^{t'}) - \log \pi(\mathbf{a^{t'}}|s^{t'})|s^{t+1}\big],$$

$$Q(s^t, \mathbf{a}^t) := \sum_{i=1}^{N} r_i(s^t, a_i^t, a_{\mathcal{N}_i}^t) + \mathbb{E}_{p(s^{t+1}|s^t, \mathbf{a}^t)}[V(s^{t+1})] \tag{14}$$

Thus $V(s^t) = E_\pi[Q(s^t, a^t) - \log \pi(\mathbf{a^t}|s^t)]$. The optimal policy $\pi^*(\mathbf{a^t}|s^t) = \frac{\exp(Q(s^t, \mathbf{a}^t)}{\int \exp Q(s^t, \mathbf{a}^t) d\mathbf{a}^t}$ by plugging the definition of $Q$ into equation 13.

Remind in section 4.1, we have approximated the optimal joint policy by the mean field approximation $\prod_{i=1}^{N} q_i(a_i|s)$. We now plug this into the definition of equation 14 and consider the discount factor. Notice it is easy to incorporate the discount factor by defining a absorbing state where each transition have $(1 - \gamma)$ probability to go to that state. Thus we have

$$V(s^{t+1}) := \mathbb{E}\big[ \sum_{t'=t+1}^{T} (\sum_{i=1}^{N} r_i(s^{t'}, a_i^{t'}, a_{\mathcal{N}_i}^{t'}) - \sum_{i=1}^{N} \log q_i(a_i^{t'}|s^{t'}))|s^{t+1}\big],$$

$$Q(s^t, \mathbf{a}^t) := \sum_{i=1}^{N} r_i(s^t, a_i^t, a_{\mathcal{N}_i}^t) + \gamma \mathbb{E}_{p(s^{t+1}|s^t, \mathbf{a}^t)}[V(s^{t+1})]. \tag{15}$$

Thus we can further decompose $V$ and $Q$ into $V_i$ and $Q_i$. We define $V_i$ and $Q_i$ in the following way.

$$V_i(s^{t+1}) = \mathbb{E}\big[ \sum_{t'=t+1}^{T} \big(r_i(s^{t'}, a_i^{t'}, a_{\mathcal{N}_i}^{t'}) - \log q_i(a_i^{t'}|s^{t'})\big)|s^{t+1}\big],$$

$$Q_i(s^t, a_i^t, a_{\mathcal{N}_i}^t) = r_i(s^t, a_i^t, a_{\mathcal{N}_i}^t) + \gamma \mathbb{E}_{p(s^{t+1}|s^t, \mathbf{a}^t)}[V_i(s^{t+1})].$$

Obviously we have $V = \sum_{i=1}^{N} V_i$ and $Q = \sum_{i=1}^{N} Q_i$.

For $V_i$, according to our definition, we obtain

$$V_i(s^t) = \mathbb{E}_{\mathbf{a}^t \sim \prod_{i=1}^{N} q_i}[r_i(s^t, a_i^t, a_{\mathcal{N}_i}^t) - \log q_i(a_i^t|s^t) + \mathbb{E}_{p(s^{t+1}|s^t, \mathbf{a}^t)}V_i(s^{t+1})]. \tag{16}$$

Now we relate it to $Q_i$, and have

$$V_i(s^t) = \mathbb{E}_{\mathbf{a}^t \sim \prod_{i=1}^{N} q_i}[Q_i(s_i^t, a_i^t, a_{\mathcal{N}_i}^t) - \log q_i(a_i^t|s^t)] = \mathbb{E}_{(a_i, a_{\mathcal{N}_i}) \sim (q_i, q_{\mathcal{N}_i})} Q_i(s_i^t, a_i^t, a_{\mathcal{N}_i}^t) - \mathbb{E}_{a_i \sim q_i} \log q_i(a_i^t|s^t).$$

Thus it suggests that we should construct the loss function on $V_i$ and $Q_i$ in the following way. In the following, we use parametric family (e.g. neural network) characterized by $\eta_i$ and $\kappa_i$ to approximate $V_i$ and $Q_i$ respectively.

$$J(\eta_i) = \mathbb{E}_{s^t \sim D}[\frac{1}{2}\big(V_{\eta_i}(s^t) - \mathbb{E}_{(a_i, a_{\mathcal{N}_i}) \sim (q_i, q_{\mathcal{N}_i})}[Q_{\kappa_i}(s^t, a_i^t, a_{\mathcal{N}_i}^t)] - \log q_i(a_i^t|s^t)\big)^2],$$

$$J(\kappa_i) = \mathbb{E}_{(s^t, a_t^t, a_{\mathcal{N}_i^t}) \sim D}[\frac{1}{2}\big(Q_{\kappa_i}^i(s^t, a_i^i, a_{\mathcal{N}_i}^t) - \hat{Q}(s^t, a_t^i, a_{\mathcal{N}_i}^t)\big)^2]. \tag{17}$$

where $\hat{Q}_i(s^t, a_i^t, a_{\mathcal{N}_i}^t) = r_i + \gamma \mathbb{E}_{s^{t+1} \sim p(s^{t+1}|s^t, a^t)}[V_{\eta_i}(s^{t+1})]$.

Now we are ready to derive the update rule of the policy, i.e., the intention propagation network.

Remind the intention propagation network actually is a mean-field approximation of the joint-policy.

$$\min_{p_1, p_2, \dots, p_n} KL(\prod_{i=1}^{N} p_i(a_i|s)||\pi^*(\mathbf{a}|s)).$$

It is the optimization over the *function* $p_i$ rather than certain parameters. We have proved that after $M$ iteration of intention propagation, we have output the nearly optimal solution $q_i$.

In the following, we will demonstrate how to update the parameter $\theta$ of the propagation network $\Lambda_\theta(\mathbf{a}^t|s^t)$, if we use neural network to approximate it. Again we minimize the KL divergence

$$\min_\theta \mathbb{E}_{s^t} KL(\prod_{i=1}^N q_{i,\theta}(a_i^t|s^t)||\pi^*(\mathbf{a}^t|s^t))$$

Plug the $\pi^*(\mathbf{a}^t|s^t) = \frac{\exp(Q(s^t,\mathbf{a}^t))}{\int \exp Q(s^t,\mathbf{a}^t)d\mathbf{a}_t}$ into the KL divergence. It is easy to see, it is equivalent to the following the optimization problem by the definition of the KL divergence.

$$\max_\theta \mathbb{E}_{s^t}\left[\mathbb{E}_{\mathbf{a}^t\sim\prod q_{i,\theta}(a_i^t|s^t)}[\sum_{i=1}^N Q_{\kappa_i}(s^t,a_i^t,a_{\mathcal{N}_i}^t) - \sum_{i=1}^N \log q_{i,\theta}(a_i^t|s^t)]\right].$$

Thus we sample state from the replay buffer and have the loss of the policy as

$$J(\theta) = \mathbb{E}_{s^t\sim D,\mathbf{a}^t\sim\prod_{i=1}^N q_{i,\theta}(a_i^t|s^t)}[\sum_{i=1}^N \log q_{i,\theta}(a_i^t|s^t) - \sum_{i=1}^N Q_{\kappa_i}(s^t,a_i^t,a_{\mathcal{N}_i}^t)].$$

