# OpenReview forum: "Intention Propagation for  Multi-agent Reinforcement Learning"
_ICLR.cc/2021/Conference — Reject_

### Official Review · AnonReviewer2 · 2020-10-22
**Official Blind Review #2**

**Rating:** 6
**Confidence:** 2

**Review:**

This paper proposes a method for generating policies in cooperative games, using a neighbourhood-based factorisation of reward, and an iterative algorithm which independently updates policies based on neighbour policies and then propagates the policy to neighbours using function space embedding.

The experimental results looked promising, so there seems to be an idea here worth communicating.

The paper was very hard for me to follow. I'm not an expert in the area and wouldn't expect to follow all of the reasoning in constructing the method, but I would be expect to be able to follow some clear statements of the algorithm, or its theoretical properties (guarantees of some solution quality given certain assumptions, the parameters affecting this, etc.). Instead the main body of the paper felt like a collection of pieces that were used when developing the algorithm. I would suggest it might be easier to follow if written from the top down, instead: present a high-level overview of the idea, give a (detailed!) description of the algorithm, the experiments, and leave the derivation to the appendix.

Despite being in the appendix, the algorithm is less than half a page, and doesn't explain the variables. eta and kappa might be described elsewhere, but it would be helpful to reference where. J is a loss: which one?

One of the claimed contributions is this is *principled* method. However, the exact assumptions are not clear, and the chain of issues discussed throughout section 4 seems to include discussion of approximation. What makes this principled? This would seem to need a clear statemen
One of the claimed contributions is this is *principled* method. However, the exact assumptions are not clear, and the chain of issues discussed throughout section 4 seems to include discussion of approximation. What makes this principled? This would seem to need a clear statement: what are the exact assumptions, and what precisely is the quality of the output? Is it exact? What are the complete set of parameters? Where does approximation fit in?
t: what are the exact assumptions, and what precisely is the quality of the output? Is it exact? What are the complete set of parameters? Where does approximation fit in?

Another claimed contribution is computational efficiency. How does the computational cost compare to the baselines in the experiments?


Proposition 1: "The optimal policy has the form ... 1/Z exp(...)"
I found the use of optimal slightly hard to follow throughout this. The usual definition of optimal policy would be a value maximising policy, which would be an argmax rather than a softmax. Following that definition, this proposition wouldn't be true, so it seems like it needs more explanation, or more careful wording.
The cited PRL article (Levine 2018) seems to retain this standard use of optimal: it uses a distribution over trajectories with an equation similar to here (a softmax over accumulated trajectory rewards), and makes use of the property that trajectories corresponding to an optimal policy have maximum probability in that distribution.
Can the authors clarify this use of optimal?

Proposition 1:
For clarity, explain the intention of psi. Is this the future accumulated reward given the current state and selected action?

=-= comments after author discussion
The authors were quite active in editing the submission, and addressing the concerns I had. I still find the paper a bit hard to follow, but none of my original concerns remain.

---

> ### Author Response · Authors · 2020-11-17
> **Responds to the review**
>
>
> Thanks for the comments.
>
> ==========Responds to the major concern.
>
> The main concern is that our paper is hard to follow. I try my best to present the roadmap of our paper that helps the reviewer to understand the paper.
>
> The motivation of this paper is to propose an algorithm to optimize the joint policy. If we directly search the  whole policy space using the brute-force, it obviously suffers from the scalability issue, i.e., exponentially large action space.  To solve such a hard problem, we use a key observation that in the real world problem the reward function has local property in a multi-agent system.
>
> Thus at the beginning of the derivation, we present an important proposition which says the optimal policy is a Markov random field (section 4.1). It reduces the exponentially large search space to a polynomial one.  But implementing a policy (e.g. sample an action) with the Markov random field form in RL  is not a  trivial problem (We call it problem A here). Therefore we propose the variational inference method, e.g., mean-field approximation or belief propagation  in the section Mean Field Approximation. However such approximation needs a complicated computation (problem B). To solve this problem, we use the tool of kernel embedding (section Embed the update rule). But the function in the formula (5) is not known (problem C). At last, we learn the function with neural networks and represent the whole algorithm.
>
>
> Basically, the structure of this paper  is clear. Our aim is to optimize the joint policy. It induces the problem A. While solving A induces problem B. Again B introduces problem C. Finally we figure out the solution of C and get the whole algorithm.  I know the reviewer would like a reversed order. However I think at least for this paper, our structure may help the reader to understand how we create this algorithm. Indeed, we have presented a lot of details of derivation in the appendix and just leave important steps in the main paper.
>
> ===============Responds to the detailed questions.
>
> 1 Concerns on the algorithm. Eta,kappa are given in section 4.2 algorithm. In fact we describe most of the content of the algorithm  related to variables and loss function in this section. The reviewer may omit this section. We move the pseudo code from the appendix to the main paper in the rebuttal version. Please check that.
>
> We called our algorithm principled method, since it builds on the theory of variational inference and optimization. It is not a heuristic algorithm, since we start from an objective function (optimize the joint point) and derive the algorithm using the tools developed by the graphical model and optimization community.
>
> In this paper, we know the quality of the solution using existing tools in graphical model and optimization. E.g. we know  the iterative rule (equation 2) of mean-field approximation converges, see details in [1],[2].  We know we can embed a distribution using RKHS. We clearly know the final leaned policy is the mean-field approximation or the loopy belief propagation of the  joint policy. The reviewer may want a sample complexity or convergence rate. I agree it will be good if we can provide that. But I think it may be too ambitious to do so at least in the current stage, since even in the single agent RL, it is still a hot research topic.
>
>
> 2 Responds to the optimal solution. Yes, we follow the definition in  the probabilistic reinforcement learning (PRL) as we mentioned in section Backgrounds. In PRL or the reinforcement learning  with entropy term, the max operator in the Bellman optimality condition would be replaced by the softmax operator. Thus the optimal policy has the softmax form. Please check the equation(6) in Soft-Q learning [3] and related work section  in Soft-Actor-Critic [4].
>
>
> 3 \Psi is similar to the Q function  but not the same. it includes some terms of the entropy on the joint policy.
>
> 4 Responds to computational efficiency. In practice, the training cost for each step of  PIC, Qmix  is larger than ours. DGN and MFQ have similar costs with us.  But our IP outperforms them a lot.  The cost  in COMA and IQL is smaller than ours since it does not need communication.  However the performance of our IP is much better.
>
>
> [1] Variational Inference: A Review for Statisticians. David Blei et al.
>
> [2] Theoretical and Computational Guarantees of Mean Field Variational Inference for Community Detection. Annals of Statistics.  Aderson Zhang and Harrision Zhou.
>
> [3] Reinforcement Learning with Deep Energy-Based Policies. Tuomas Haarnoja,  Aurick Zhou,  Pieter Abbeel,  Sergey Levine.
>
> [4] Soft Actor-Critic. Tuomas Haarnoja,  Aurick Zhou,Pieter Abbeel,  Sergey Levine

---

> > ### Comment · AnonReviewer2 · 2020-11-18
> > **re: response**
> >
> > Regarding clarity ---
> >
> > I appreciate the author's re-ordering of the paper, moving the algorithm and more discussion of it into the main body. That does help.
> >
> > The response suggests that the intention behind the current layout is showing how the authors created the algorithm. This exactly reflects my stated concerns in the initial review. Put more plainly: my first interest is the new things you are introducing, not how you came to discover/prove/write them. Emphasising the meandering road to the contributions introduces many intermediate problems, keeps changing the current objective throughout the discussion, and muddies the conclusions.
> >
> > ----------
> > Optimal policy ---
> >
> > I did read the cited work by Levine 2018, and this is part of where my concern came from. In that work, there is a clear distinction between the optimal policy and a soft-max policy based on rewards.
> >
> > Levine 2018 is fairly careful to keep that distinction clear, after making a model assumption of setting the probability p(O_t=1|s_t,a_t) of time step t being optimal to be exponentiated rewards. For example, phrases like "how can we formulate a probabilistic graphical model such that the most probable trajectory corresponds to the trajectory from the optimal policy?" make clear the distinction between model-optimal and optimal.
> >
> > That assumption and distinction seem to be missing in this work. Even if it's common to make that model assumption, and/or assume softmax-reward-policy is "optimal", it's worth taking at least one sentence to make clear that these assumptions are being made.
> >
> > Given this assumption and the various approximation steps made, the paper claim of "while our work learns the optimal policy" should maybe be better phrased as "while our work aims to learn the optimal policy".
> >
> > ----------
> > Algorithm and description of variables and functions ---
> >
> > I could have been more clear by stating "eta and kapp *are* described elsewhere" rather than saying "might be". My intention was to say that the description of the algorithm should be self-contained. Even if parts were described elsewhere, it would be helpful to include them again with the algorithm.
> >
> > I think this has been addressed by moving the algorithm into the main body of the text.
> >
> > ----------
> > Principled ---
> >
> > Maybe a minor point, and just a different use of language. However,  I would expect a claim of principled to include some guarantees, or at least make the assumptions made clear, and hopefully give some idea about the consequences of those assumptions. There seem to be a number of steps in the chain of reasoning which approximate the optimal answer. By how much? Under what circumstances? The paper didn't make that clear to me, which makes the distinction of "principled" vs. prior work a bit hard to agree with.

---

> > > ### Author Response · Authors · 2020-11-21
> > > **Responds to the new comments**
> > >
> > > Thanks for the useful suggestion from the reviewer.
> > >
> > > To the question on the optimality.  We emphasize the assumption of the optimal policy  and the softmax form of the optimal policy in the section of backgrounds.  We also add some words to explain that our results follow the definition of the optimal policy in (Levine 2018 ) before proposition 1. Please check the  new version.
> > >
> > > To the question on the Principled method. We clarify the assumption of kernel embedding method in the section of backgrounds and why its holds.  We also add some discussions in the sections of Mean field approximation and  Embed   the   update   rule  to give some reason why our algorithm converges to the mean field solution even under some approximations. Please check the new version of this paper.

---

> > > > ### Comment · AnonReviewer2 · 2020-11-22
> > > > **response**
> > > >
> > > > I still find the paper a bit hard to follow, but the steps where assumptions are made now seem to be mentioned.
> > > >
> > > > Thank you for the revisions.

---

### Official Review · AnonReviewer4 · 2020-10-27
**Interesting theoretical insight, limited algorithmic novelty**

**Rating:** 7
**Confidence:** 4

**Review:**

Paper Summary
The paper considers the cooperative multiagent MARL setting where each agent’s reward depends on the state and the actions of itself and its neighbors The paper has a theoretical claim that, for such reward structure, the optimal maximum entropy joint policy in the form that can be factored into potential functions, one for each agent. In particular, if the sum of all agents’ rewards is a function on pairwise actions, those potential functions are one for each agent and one for each pair of actions (i.e. the equation after Proposition 1).
Then, the paper proposes to use mean-field approximation to approximate the optimal joint policy (Equation (3)), which leads to a concrete algorithm that relies on passing the embedding of each agent’s local policy around to neighbors. The paper then empirically shows that the algorithm is particularly effective for domains with a large number of agents.


Major Comments/Questions
1. Although the motivation has an interpretation of intention propagation, the resulting architecture (Figure 1b) and loss functions (Section 4.2) seems to be a standard messaging passing architecture with SAC loss functions that loses the *intention* semantics. I do not see too much algorithmic novelty here.

2. For the baselines used in the experiments, it seems that only IP and DGN allow communication/message passing *during execution*, which makes it unsurprising that the two methods outperform other baselines.

Minor Comments/Questions
1. The beginning of Section 3 says the paper considers maximum entropy as the optimization objective, while eta(pi) at the beginning of Section 4 says the objective is long-term reward (no entropy). This seems to be an inconsistency here.

2. For the assumptions on rewards, Proposition 1 assumes that each agent’s reward depends on its neighbors, while the derivation of Equation (3) (and thus the following algorithm) further assumes that the reward depends on pairwise actions. It is a little bit unclear what assumptions are required for all the theoretical and experimental claims of this paper.

3. Is there reason to believe that the multi-round message passing will converge to the fixed-point of Equation (2)?

4. What is the "overgeneralization issue"?


Overall (weak accept)
The paper has a clear introduction and motivation of the proposed algorithm. The insight that optimal maximum entropy joint policy takes the format of Markov Random Field might be of some value and interest. However, I don’t think the resulting method has much algorithmic novelty.


--------------
Thanks for the response and I've increased my score. I am satisfied with the response but still not convinced about the algorithmic novelty on the intention semantics built into the method, even after reading B.1.  In particular, it seems that the loss functions do not drive mu's represented by NNs to the fixed point solution of Eq (3); psi shows up in Eq (3) but does not play a role in the following development of the method.

---

> ### Author Response · Authors · 2020-11-17
> **Responds to the review**
>
> Thanks for the comments.
>
> ===========Response to major comments.
>
> Q1. Question on the baselines of  the communication during execution. MFQ is another baseline with communication during execution. Each agent’s policy in MFQ  needs the information of mean action of other agents.  We have discussed the difference between our work and MFQ in the introductions.
>
> Also notice in the implementation of PIC (please check the code released by the author of PIC), each actor concatenates the state information from other agents to get a global state. However all of our  experiments are partially observed. Thus you can think that  each agent obtains the state information of other agents through communication during the execution.
>
> Therefore, MFQ and PIC are another two baselines with communication during execution.
>
>
>
>
> Q2. Question on the algorithmic novelty. The architecture in Figure 1 (b) is just one simplest instance derived by our theoretical framework. We choose it to illustrate in the main paper since it is easy for the reader to understand. In fact, we have explained that this framework can generate  a series of network structures, e.g., the formula in appendix B.2. They are not standard message passing architecture. We have emphasized this in remarks in page 2 and the end of section 4.1.  The reviewer may miss that. We also explained Figure 1 (b) in the semantic of intention in the appendix B.1. We would move these two parts to the main paper, if space allows.
>
>
> ==========Response to minor comments.
>
> Good comments on inconsistency of objective function. We  rectify the objective function in section 4 and add an entropy term in the new version.
>
> In general, we do not need the assumption of pairwise action. Since the mean-field approximation in the probabilistic graphical model does not require the pairwise assumption. We just want to ease the exposition here.
>
> Such updates will converge to the fixed points.  We add some discussions in the rebuttal version.  Please check that. For related theoretical results, please see the detailed discussion in the following reference.
> Variational Inference: A Review for Statisticians. David Blei et al.
>
> Theoretical and Computational Guarantees of Mean Field Variational Inference for Community Detection. Annals of Statistics.  Aderson Zhang and Harrision Zhou.
>
> A Hilbert space embedding for distributions. A. Smola, A. Gretton, L. Song, B. Schölkopf.
>
> The definition of  relative overgeneralization is related to game theory. Please check following reference
>
>  Lenient learning in independent learner stochastic cooperative games. Wei et al, JMLR 2016
> The Representational Capacity of Action-Value Networks for Multi-Agent Reinforcement Learning  Castellini et al. AAMAS  2019.

---

### Official Review · AnonReviewer1 · 2020-10-29
**Promising approach using mean-field estimates to learn joint policy with suitable set of experiments**

**Rating:** 6
**Confidence:** 4

**Review:**

The paper proposes a scalable approach via intention propagation to learn a multi-agent RL algorithm using communication in a structured environment. An agent encodes its policy and sends the “intention” to the neighboring agents with the assumption that only the closest agents would be the affected by it. The approach involves using techniques from the embedded probabilistic inference literature using mean-field variational inference. The joint-policy is estimated using the mean-field approximation that is obtained via propagating intents in an iterative manner. So this approach helps in avoiding the need to factorize the value function explicitly.

The related works section does a nice survey of related approaches and the paper shows conceptual differences to an earlier proposed MFG that has stricter requirements.

The experiments shown cover many important baselines that are shown to be good baselines in respective environments. IP outperforms all the baselines in three competitive benchmarks.

I have a few questions about the clarity of the presentation.

- How important is the graph structure defined by k-means? A comparison with a randomized graph and ablation with different reset time (n) intervals would be interesting.

- In the experiments, it would be interesting to check if intention only helps the nearby agents. How does adding/removing agents to the set of neighbors affect learning? A comparison with a fully connected graph should be sufficient. The plot in the Appendix shows results on the CityFlow task which has very structured observation with the set of immediate neighbors always set of 4. Doing such an analysis on a more dynamic environment like MPE would be helpful.

- What is the computational cost of a densely connected graph as compared to method without using a fixed topology?

- Fig 4c does not show plots until convergence.

Overall I feel some restructuring of the paper would benefit the reader explaining some missing portions of the algorithm. For eg, taking out the environment images from the main text.

---

> ### Author Response · Authors · 2020-11-17
> **Responds to the comment**
>
> Thanks for the comments.
>
> Q1 “How important is the graph structure defined by k-means? A comparison ...”
> We add two results in appendix F.2 in the rebuttal version of this paper. Please check Figure 8 (e) and (f) . Figure 8 (e) says that the K-nearest neighbor graph structure is important. In Figure 8 (f), we update the adjacency matrix every 5 or 10 steps. In general, the performance decreases.
>
> Q2 “Doing such an analysis on a more dynamic environment like MPE would be helpful.”
> We have done the ablation study on the number of neighbors in Cooperative navigation (dynamic environment) in Figure 8c in appendix.  In that figure neighbors =30 denotes a fully connected graph (there are 30 agents).  Ip with Neighbors =8 has the best performance. The reviewer may want to check that result.
>
> Q3 “What is the computational cost of a densely connected graph as compared to a method without using a fixed topology?”
> In the dynamic environment, we need to update a sparse adjacency matrix and it is basically a simple K nearest neighbor problem. In general K is a small number. It is just a very small overhead in practice and  can be accelerated further using other techniques, e.g., faiss by facebook.  We also can update the adjacency matrix every n steps to save the computation. We do a new ablation study in Figure 8 (f) in the rebuttal  version.
>
> However if the graph is densely (fully) connected, each agent needs to aggregate all other agent’s information, which in practice causes the heavy burden on the training and memory usage (since the size of the weight matrix becomes large). Roughly speaking, in our experiment, the time cost in updating the adjacency matrix is less than 1/5 of total time.  However If we use the fully connected graph, the training time will be 2 or 3 times slower.
>
> ========Responds to other questions.
>
> I  move the algorithm in the appendix to the main paper in the rebuttal version to help the reader to understand the paper. Please check that.
>
> We run experiments in Figure 4(c) longer and update it in the rebuttal version of this paper.  In this experiment, the asymptotic results of PIC and IP are similar, but IP converges faster.  Notice IP is still much better than PIC in other experiments.

---

### Official Review · AnonReviewer5 · 2020-11-07

**Rating:** 5
**Confidence:** 4

**Review:**

The paper proposes a multi-hop communication method for multi-agent reinforcement learning. This method is based on the loosely coupled reward structures among agents, which, as far as I am concerned, are generally held in complex multi-agent settings. The authors use experiments on CityFlow, MPE, and MAgent to demonstrate that their method can outperform the SoTA methods and is scalable. The empirical results is impressive. However, it is some concerns regarding methods that lead to my overall negative rating.

Firstly, also the most importantly. Although the authors emphasize that they are communicating the intentions of agents, I think their method is quite similar to those communicating local observations, like NDQ (https://arxiv.org/abs/1910.05366), DGN, or CollaQ (https://arxiv.org/abs/2010.08531). One way to interpret the proposed communicating network structure is a normal multi-hop communication mechanism, but only with a softmax activation function.

Compared to previous works studying communications of local observations, the proposed work (1) needs to address the problems induced by the joint policy, like sampling from it. The author use a variational influence approach to conduct sampling. However, this approach may hurt the scalability. And it (2) requires agents have access to the global states. For partial observable environments, the proposed methods needs to reply on DGN.

Some other points: (1) I was expecting ablation studies where DGN is ablated on partial observable environments. (2) Some parts in the method section are hard to follow.

---

> ### Author Response · Authors · 2020-11-17
> **Responds to the review**
>
> Thanks for the comments.
>
> ============Responds to the main concern.
>
> The main concern of the reviewer  5 is that our method is  similar to NDQ, DGN, CollaQ. However our method is quite different.  I also notice CollaQ is currently under review at ICLR2021! It is unfair for us to discuss the relation to that paper. In the following I discuss the detailed difference from NDQ and DGN.
>
> 1.   We provide a framework inspired by theory. Figure 1 is just the simplest instance of this framework!  The aim is to ease the understanding for readers.  This framework can be instantiated to several different structures, as we emphasized in remarks in page 2. The message passing formula corresponding to loopy belief propagation is given in appendix B.2.   That formula is quite different from the vanilla message passing  work in NDQ and DGN.
>
> 2.   Even if we constrain our discussion on the structure in Figure 1, our message passing is quite different from that in DGN and NDQ. Note that two key questions in message passing of MARL are “what to communicate” and “how the agent reacts to that message”. We will discuss the difference from these two aspects.
>
>      In fact, we have discussed differences from  DGN （Jiang et al 2020） in the related work section (we emphasize that by red
>      color in the rebuttal version), the reviewer may omit that.  DGN  sends the state embedding. The aim is to mitigate the partially
>      observed problem. However, each agent still behaves independently (it learns Q(s,a_1), Q(s,a_2),..Q(s,a_n) rather than joint Q)
>     ,i.e., factorized policy. Our work aims to learn the joint policy pi(s,a_1,a_2,...,a_n) (the motivation of this paper) and it sends the
>     embedding of the policy. In addition, we prove that following our update rule in equation (5)  (That is how the agent reacts to
>     the received message ) , the learned policy is a mean field approximation or belief propagation of the optimal policy. The
>     difference is also quite clear in the empirical study (see the Experiment ). See our discussion in the Section of Stability in the  experiment (page 9 in the rebuttal version.  page 8 in the original version) .
>
>     The motivation of NDQ is to learn a nearly decomposable Q function via communication minimization. NDQ is builded on Qmix
>     but adds a regularization to maximize mutual information between message and agent’s action selection while minimizing the
>     entropy of the message. The structure in their Figure 1 does not have any similarity to the graph (neural network) structure in
>     our Figure 1 or the formula in Appendix B.2.
>
>    In addition, In NDQ, the content of the message is to “reduce the uncertainty in action-value function” (see page 3 in NDQ).
>    While our one is to approximate the joint policy. Therefore the object function of these two works are fundamentally different.
>
>
>
> 3.  “The author use a variational influence approach to conduct sampling. However, this approach may hurt the scalability”. Do you mean that our method has a scalability issue?  Our method does not have such issue. The fact is that our method can support  1225 agents setting  in the experiment. It outperforms other baselines especially in large scale settings. In practice, the training time for each step is almost the same with other methods with communication.
>
> 4. "It requires agents have access to the global states. For partial observable environments, the proposed methods needs to reply on DGN." Our paper does not rely on DGN.  The only similarity is that DGN and IP need to send the state embedding to other agents in partial observable setting.
>
> ==============Responds to some minor point.
>
>
> 1. The reviewer says “I was expecting ablation studies where DGN is ablated on partial observable environments”. Do you want an ablation study on DGN or IP(our method)? If it is IP, we have already done that in the ablation study on the number of neighbors and graphs (Figure 8 (a) (b) (c))  in the section F of appendix.  In these ablation studies, if we use the fully connected graph (fully connected graph in cityflow and N=30 in cooperative navigation),  you can think that  the agent can obtain global information.  The different numbers of neighbors reflect  different degree of the partial observation.
>
>     In fact,  all experiments in the paper are tested in the partially observed setting. We emphasize this in the section 5.1. Please check the new version.
>
> 2. We add some discussions in the method section to help the reader  to understand the method. Please check the new version.

---

> > ### Comment · AnonReviewer5 · 2020-11-23
> > **Re: Response**
> >
> > I want to thank the reviewer for the response.
> >
> > My stress is not that your method has a scalability issue. I had concerns about using variation inference for sampling from the joint policy. Would it be accurate?
> >
> > In the paper, authors state that "We use the graph neural network on state of agent to gradually increase the receptive field of states on each agent as that in (Jiang et al., 2020)". This raises my concern about the dependence on DCG. Did you mean that the intention propagation can not include enough information about local observations and you have to conduct two means of communication (one for intention propagation, one for local information sharing)? If so, which part contributes more to the performance?

---

> > > ### Author Response · Authors · 2020-11-23
> > > **Responds to the new comment**
> > >
> > > Thanks for the comments.
> > >
> > > 1. Good question on the variational inference. The answer is that the variational inference is accurate enough in both practice and theory.
> > >
> > >     In practice, variational inference method is a  classical method to learn the  Markov Random Field (recall we prove the optimal
> > >     joint policy is a Markov Random Field in proposition 1 ) model [1]. It has been tested on computer vision, language model, and
> > >    many others[2][3]  and shows good performance.
> > >
> > >     For the theoretical guarantee on the variational inference, the reviewer may want to check [2][4]. We also  give some discussion
> > >     on why our IP converges to the fixed point of equation (2) in the rebuttal version.
> > >
> > >    [1] An Introduction to Variational Methods for Graphical Models. Michael Jordan, Zoubin Ghahramani, Tommi  jaakkola,
> > >    Lawrence K. Saul.
> > >    [2] Graphical Models, Exponential Families, and Variational Inference. Martin Wainwright, Michael Jordan
> > >    [3] Markov random field modeling in image analysis.  Stan Z. Li
> > >    [4] Theoretical and Computational Guarantees of Mean Field Variational Inference for Community Detection. Annals of Statistics.
> > >    Aderson Zhang and Harrision Zhou.
> > >
> > >
> > >
> > > 2. Good question on the communication.
> > >
> > >    Yes, we have two means of communication.  The word "we use the graph neural network..." may cause some misunderstanding
> > >   for the reviewer. Sorry for that. I rectify it in the rebuttal version.
> > >
> > >   For the contribution of these two parts. In general, in the large scale setting, the contribution of intention propagation is larger
> > >  (in the small scale setting, the performance of all baselines are similar.).
> > >
> > >    We can compare the gap between  IQL (independent Q learning ) and DGN   with the gap between DGN and IP, Since DGN
> > >    introduces the local information sharing (on state) compared with IQL.  While our IP contains both local information sharing and
> > >   intention propagation.
> > >
> > >   In Figure 3 b, the gap between  IQL and DGN is about 600. The gap between DGN and IP is about 700.  In Figure 3 c, the gap
> > >  between  IQL and DGN is about 2e3 while the gap between DGN and IP is about 1e4!
> > >
> > >  Thus intention propagation brings  a large performance gain  especially in the large scale setting. Similar result holds in Figure 4 (see that the gap between DGN and IP in Figure 4(c) is very large)  and the result in appendix. Please check that.

---

> > > ### Author Response · Authors · 2020-11-24
> > > **Looking forward to feedback on new responds.**
> > >
> > > Dear reviewer,
> > >
> > > We are wondering if our new responses have cleared all your concerns including the two questions yesterday. We would appreciate it if you could re-evaluate our submission, or kindly let us know whether you have any other questions, so that we have time to respond and improve our manuscript in the remaining several hours of the rebuttal period. Thanks a lot.
> > >
> > > Best regards,
> > >
> > > The Authors

---

### Decision · Program_Chairs · 2021-01-07
**Final Decision**

**Decision:**

Reject

**Comment:**

The paper describes a framework for multi-agent reinforcement learning that uses Markov Random Fields. Unfortunately, the paper is not clearly written and would benefit from significant revisions that improve its structure and make the model and approximations more explicit.

In particular, the paper says a graph says which agents $i,j$ communicate. This is typically called the "coordination graph" in this setting, see
"Collaborative Multiagent Reinforcement Learning by Payoff Propagation", Kok and Vlassis, 2006. Note that within that paper they provide Q-function decomposition, which can only serve to approximate the optimal policy.

The authors of this submission claim that an MRF is sufficient for optimal policies. I fail to see how this is true. In particular, Proposition 1 has to be checked more carefully. I tried to go through it, but it did not seem to make sense to me. Why is there an exp() term in the definitoin of the optimal trajectory probability? Why would minimising the KL divergence be enough to obtain an optimal policy? Perhaps it gives an optimal policy within the class of MRF policies, but that's not the same thing as the globally optimal policy.

Overall, I find the lack of clarity and in depth discussion of early related work disturbing, particularly with respect to the theoretical claims in the paper.